# Knowledge Diffusion for Distillation

**Tao Huang**[1,2]    **Yuan Zhang**[3]    **Mingkai Zheng**[1]    **Shan You**[2*]
**Fei Wang**[4]    **Chen Qian**[2]    **Chang Xu**[1]
[1]School of Computer Science, Faculty of Engineering, The University of Sydney
[2]SenseTime Research    [3]Peking University
[4]University of Science and Technology of China

## Abstract

The representation gap between teacher and student is an emerging topic in knowledge distillation (KD). To reduce the gap and improve the performance, current methods often resort to complicated training schemes, loss functions, and feature alignments, which are task-specific and feature-specific. In this paper, we state that the essence of these methods is to discard the noisy information and distill the valuable information in the feature, and propose a novel KD method dubbed DiffKD, to explicitly denoise and match features using diffusion models. Our approach is based on the observation that student features typically contain more noises than teacher features due to the smaller capacity of student model. To address this, we propose to denoise student features using a diffusion model trained by teacher features. This allows us to perform better distillation between the refined clean feature and teacher feature. Additionally, we introduce a light-weight diffusion model with a linear autoencoder to reduce the computation cost and an adaptive noise matching module to improve the denoising performance. Extensive experiments demonstrate that DiffKD is effective across various types of features and achieves state-of-the-art performance consistently on image classification, object detection, and semantic segmentation tasks. Code is available at https://github.com/hunto/DiffKD.

## 1   Introduction

The success of deep neural networks is generally accomplished with the requirements of large computation and memory, which restricts their applications on resource-limited devices. One widely-used solution is knowledge distillation (KD) [14], which aims to boost the performance of efficient model (student) by transferring the knowledge of a larger model (teacher).

The key to knowledge distillation lies in how to transfer the knowledge from teacher to student by matching the output features (*e.g.*, representations and logits). Recently, some studies [17, 29] have shown that the discrepancy between student feature and teacher feature can be significantly large due to the capacity gap between the two models. Directly aligning those mismatched features would even disturb the optimization of student and weaken the performance. As a result, the essence of most state-of-the-art KD methods is to shrink this discrepancy and only select the valuable information for distillation. For example, TAKD [29] introduces multiple middle-sized teach assistant models to bridge the gap; SFTN [30] learns a student-friendly teacher by regularizing the teacher training with student; DIST [17] relaxes the exact matching of teacher and student features of Kullback-Leibler (KL) divergence loss by proposing a correlation-based loss; MasKD [18] distills the valuable information in the features and ignores the noisy regions by learning to identify receptive regions that contribute to the task precision. However, these methods need to resort to either complicated training schemes or task-specific priors, making them challenging to apply to various tasks and feature types.

---

*Correspondence to: Shan You <`youshan@sensetime.com`>.

37th Conference on Neural Information Processing Systems (NeurIPS 2023).

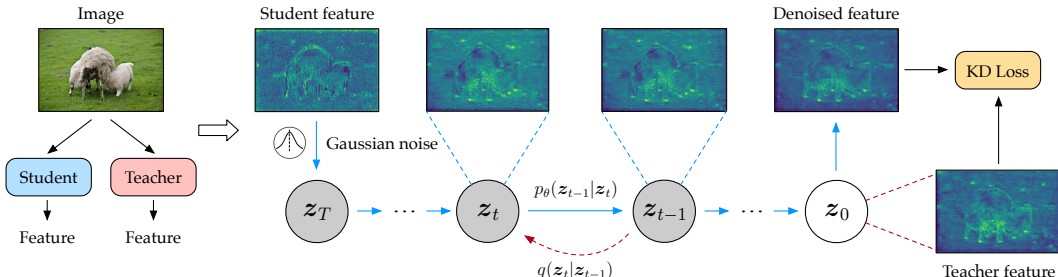

Figure 1: **Diffusion model in DiffKD.** The diffusion model is trained with teacher feature in diffusion process $q$ (red-dashed arrow), while we feed the student feature to the reverse denoising process $p_\theta$ (blue arrows) to obtain a denoised feature for distillation. We find that due to capacity limitation, student feature contains more noises and its semantic information is not as salient as the teacher's. Therefore, we treat student feature as a noisy version of teacher feature, and propose to denoise student feature using a diffusion model trained with teacher feature.

In this paper, we proceed from a different perspective and argue that the devil of knowledge distillation is in the noise within the distillation features. Intuitively, we regard the student as a *noisy* version of the teacher due to its limited capacity or training recipe to learn truly valuable and decent features. However, distilling knowledge with this noise can be detrimental for the student, and may even lead to undesired degradation. Therefore, we propose to eliminate the noisy information within student and distill only the valuable information accordingly. Concretely, inspired by the success of generative tasks, we leverage diffusion models [15, 40], a class of probabilistic generative models that can gradually remove the noise from an image or a feature, to perform the denoising module. An overview of our DiffKD is illustrated in Fig. 1. We empirically show that this simple denoising process can generate a denoised student feature that is very similar to the corresponding teacher feature, ensuring that our distillation can be performed in a more consistent manner.

Nevertheless, directly leveraging diffusion models in knowledge distillation has two major issues. (1) *Expensive computation cost.* The conventional diffusion models use a UNet-based architecture to predict the noise, and take a large amount of computations to generate high-quality images[2]. In DiffKD, a lighter diffusion model would suffice since we only need to denoise the student feature. We therefore propose a light-weight diffusion model consisting of two bottleneck blocks in ResNet [11]. Besides, inspired by Latent Diffusion [35], we also adopt a linear autoencoder to compress the teacher feature, which further reduces the computation cost. (2) *Inexact noisy level of student feature.* The reverse denoising process in diffusion requires to start from a certain initial timestep, but in DiffKD, the student feature is used as the initial noisy feature and we cannot directly get its corresponding noisy level (timestep); therefore, the inexact noisy level would weaken the denoising performance. To solve this problem, we propose an adaptive noise matching module, which measures the noisy level of each student feature adaptively and specifies a corresponding Gaussian noise to the feature to match the correct noisy level in initialization. With these two improvements, our resulting method DiffKD is efficient and effective, and can be easily implemented on various tasks.

It is worth noting that one of the merits of our method DiffKD is feature-agnostic, and the knowledge diffusion can be applied to different types of features including intermediate feature, classification output, and regression output. Extensive experimental results show our DiffKD surpasses current state-of-the-art methods consistently on standard model settings of image classification, object detection, and semantic segmentation tasks. For instance, DiffKD obtains 73.62% accuracy with MobileNetV1 student and ResNet-50 teacher on ImageNet, surpassing DKD [53] by 1.57%; while on semantic segmentation, DiffKD outperforms MasKD [18] by 1% with PSPNet-R18 student on Cityscapes test set. Moreover, to demonstrate our efficacy in eliminating discrepancy between teacher and student features, we also implement DiffKD on stronger teacher settings that have much more advanced teacher models, and our method significantly outperforms existing methods. For example, with Swin-T student and Swin-L teacher, our DiffKD achieves remarkable 82.5% accuracy on ImageNet, improving KD baseline with a large margin of 1%.

---

[2]For example, the diffusion model in Stable Diffusion [35] takes 104 GFLOPs on an input resolution of 256 × 256 [33].

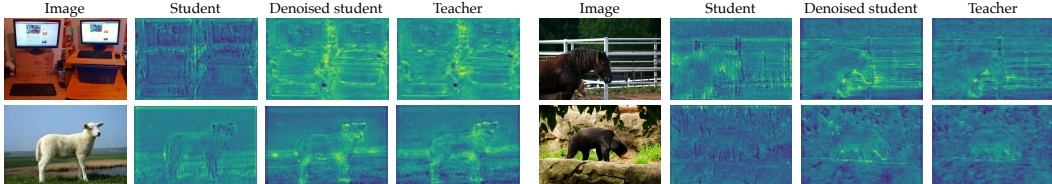

Figure 2: **Visualization of student features, denoised student features and teacher features on COCO dataset.** Details and more visualizations can be found in Appendix F.

## 2 Preliminaries

### 2.1 Knowledge distillation

Conventional knowledge distillation methods transfer the knowledge of a pretrained and fixed teacher model to a student model by minimizing the discrepancies between teacher and student outputs. Typically, the outputs can be the predictions (*e.g.*, logits in classification task) and intermediate features of model. Given the student outputs $\boldsymbol{F}^{(s)}$ and teacher outputs $\boldsymbol{F}^{(t)}$, the knowledge distillation loss is defined as

$$\mathcal{L}_{\mathrm{kd}} := d(\boldsymbol{F}^{(s)}, \boldsymbol{F}^{(t)}), \tag{1}$$

where $d$ denotes distance function that measures the discrepancy of two outputs. For example, we can use Kullback–Leibler (KL) divergence for probabilistic outputs, and mean square error (MSE) for intermediate features and regression outputs.

### 2.2 Diffusion models

Diffusion models are a class of probabilistic generative models that progressively add noise to the sample data, and then learn to reverse this process by predicting and removing the noise. Formally, given the sample data $\boldsymbol{z}_0 \in \mathbb{R}^{C \times H \times W}$, the forward noise process iteratively adds Gaussian noise to it, *i.e.*,

$$q(\boldsymbol{z}_t|\boldsymbol{z}_0) := \mathcal{N}(\boldsymbol{z}_t|\sqrt{\bar{\alpha}_t}\boldsymbol{z}_0, (1 - \bar{\alpha}_t)\boldsymbol{I}), \tag{2}$$

where $\boldsymbol{z}_t$ is the transformed noisy data at timestep $t \in \{0, 1, ..., T\}$, $\bar{\alpha}_t := \Pi_{s=0}^t \alpha_s = \Pi_{s=0}^t (1 - \beta_s)$ is a notation for directly sampling $\boldsymbol{z}_t$ at arbitrary timestep with noise variance schedule $\beta$ [15]. Therefore, we can express $\boldsymbol{z}_t$ as a linear combination of $\boldsymbol{z}_0$ and noise variable $\boldsymbol{\epsilon}_t$:

$$\boldsymbol{z}_t = \sqrt{\bar{\alpha}_t}\boldsymbol{z}_0 + \sqrt{1 - \bar{\alpha}_t}\boldsymbol{\epsilon}_t, \tag{3}$$

where $\boldsymbol{\epsilon}_t \in \mathcal{N}(\boldsymbol{0}, \boldsymbol{I})$. During training, a neural network $\Phi_\theta(\boldsymbol{z}_t, t)$ is trained to predict the noise in $\boldsymbol{z}_t$ w.r.t. $\boldsymbol{z}_0$ by minimizing the L2 loss between them, *i.e.*,

$$\mathcal{L}_{\mathrm{diff}} := ||\boldsymbol{\epsilon}_t - \Phi_\theta(\boldsymbol{z}_t, t)||_2^2. \tag{4}$$

During inference, with the initial noise $\boldsymbol{z}_t$, the data sample $\boldsymbol{z}_0$ is reconstructed with an iterative denoising process using the trained network $\Phi_\theta$:

$$p_\theta(\boldsymbol{z}_{t-1}|\boldsymbol{z}_t) := \mathcal{N}(\boldsymbol{z}_{t-1}; \Phi_\theta(\boldsymbol{z}_t, t), \sigma_t^2 \boldsymbol{I}), \tag{5}$$

where $\sigma_t^2$ denotes the transition variance in DDIM [40], which accelerates denoising process by sampling with a small number of score function evaluations (NFEs), *i.e.*, $\boldsymbol{z}_T \to \boldsymbol{z}_{T-\Delta} \to ... \to \boldsymbol{z}_0$, where $\Delta$ is the sampling interval.

In this paper, we leverage a diffusion model to eliminate the noises in student feature in our knowledge distillation method DiffKD, which will be introduced in the next section.

## 3 Method

In this section, we formulate our proposed knowledge distillation method DiffKD. We first convert the feature alignment task in KD to the denoising procedure in diffusion models, this enables us to use diffusion models to match student and teacher outputs for more accurate and effective distillation.

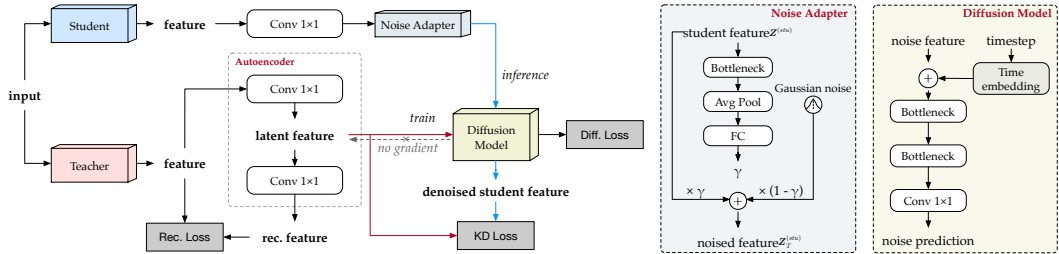

Figure 3: **Architecture of DiffKD.** *Bottneck* denotes the Bottneck block in ResNet [11].

To further improve the computation efficiency, we introduce a feature autoencoder to reduce the dimensions of feature maps, thereby streamlining the diffusion process. Additionally, we propose an adaptive noise matching module to enhance the denoising performance of student feature. The architecture of DiffKD is illustrated in Fig. 3.

## 3.1 Knowledge diffusion for distillation

Generally, models with different capacities and architectures produce varying preferences on feature representations, even when trained on the same dataset. This discrepancy between teacher and student is crucial to the success of knowledge distillation. Previous studies [19, 21] have investigated the differences between teacher and student features. Kundu et al. [19] observes that the predicted probabilistic distribution of teacher is more sharp and confident than the student's. Similarly, ATS [21] discovers that the variance of wrong class probabilities of teacher is smaller than that of student, indicating that the teacher output is cleaner and more salient. In summary, both studies find that the student has larger values and variances on wrong classes than the teacher, suggesting that the predictions of student contains more noises than the teacher's.

In this paper, we demonstrate that the same trend holds for intermediate features. We visualize the first feature map of FPN in RetinaNet [23] on COCO dataset [24] in Fig. 2 and find that the semantic information in teacher feature is much more salient than the student feature. Therefore, we can conclude that both predictions and intermediate features of student model contain more noises than the teacher's, and these noises are difficult to eliminate through simple imitation of the teacher model in KD due to the capacity gap [17]. As a result, a proper solution is to disregard the noises and only imitate the valuable information from both teacher and student. Inspired by the success of eliminating noises in diffusion models [15, 35, 40], we propose to treat the student feature as a noisy version of teacher feature, and train a diffusion model with teacher feature, then use it to denoise the student feature. With the denoised feature that contains only valuable information as teacher feature, we can perform noiseless distillation on them.

Formally, with teacher feature $\boldsymbol{F}^{(tea)}$ and student feature $\boldsymbol{F}^{(stu)}$ used in distillation, we use $\boldsymbol{F}^{(tea)}$ in the forward noise process $q(\boldsymbol{F}_t^{(tea)}|\boldsymbol{F}^{(tea)})$ (Eq. (2)) to train the diffusion model with $\mathcal{L}_{\mathrm{diff}}$ (Eq. (4)). Then the student feature is fed into the iterative denoising process of the learned diffusion model, *i.e.*, $p_\theta(\boldsymbol{F}_{t-1}^{(stu)}|\boldsymbol{F}_t^{(stu)})$ in Eq. (5), where $\boldsymbol{F}^{(stu)}$ is the initial noisy feature of the process. After this denoising process, we obtain a denoised student feature $\hat{\boldsymbol{F}}^{(stu)}$, which is used to compute the KD loss with the original teacher feature $\boldsymbol{F}^{(tea)}$ in Eq. (1).

## 3.2 Efficient diffusion model with linear autoencoder

However, we find that the denoising process in DiffKD can be computationally expensive due to the large dimensions of the teacher feature. During training, DiffKD needs to forward the noise prediction network $\Phi_\theta$ for $T$ times (we use $T = 5$ in our method) for denoising the student feature and 1 time for training the noise prediction network with teacher feature. This $T + 1$ times of forwarding can result in a high computation cost when the dimension of teacher feature is large. To reduce the computation cost of diffusion model, we propose a light diffusion model which is stacked with two bottleneck block in ResNet [11]. Then we follow Latent Diffusion Models [35] and propose to compress the number of channels using a linear autoencoder module. The compressed latent feature is used as the input of diffusion model. As shown in Fig. 3, our linear autoencoder module is composed of two convolutions only, one is the encoder for reducing the number of channels, another one is the decoder

Table 1: **Training strategies on image classification tasks.** *BS*: batch size; *LR*: learning rate; *WD*: weight decay; *LS*: label smoothing; *EMA*: model exponential moving average; *RA*: RandAugment [7]; *RE*: random erasing; *CJ*: color jitter.

| Strategy | Dataset | Epochs | Total BS | Initial LR | Optimizer | WD | LS | EMA | LR scheduler | Data augmentation |
|---|---|---|---|---|---|---|---|---|---|---|
| A1 | CIFAR-100 | 240 | 64 | 0.05 | SGD | $5 \times 10^{-4}$ | - | - | $\times 0.1$ at 150,180,210 epochs | crop + flip |
| B1 | ImageNet | 100 | 256 | 0.1 | SGD | $1 \times 10^{-4}$ | - | - | $\times 0.1$ every 30 epochs | crop + flip |
| B2 | ImageNet | 450 | 768 | 0.048 | RMSProp | $1 \times 10^{-5}$ | 0.1 | 0.9999 | $\times 0.97$ every 2.4 epochs | {*B1*} + RA + RE |
| B3 | ImageNet | 300 | 1024 | 5e-4 | AdamW | $5 \times 10^{-2}$ | 0.1 | - | cosine | {*B2*} + CJ + Mixup + CutMix |

for reconstruct the teacher features. The output feature of encoder is used to train the diffusion models (Eq. (2)) and supervise the student.

The autoencoder is trained with a reconstruction loss only, which is the mean square error between the original teacher teacher $\boldsymbol{F}^{(tea)}$ and reconstructed teacher feature $\tilde{\boldsymbol{F}}_{ae}^{(tea)}$, *i.e.*,

$$\mathcal{L}_{\text{ae}} := ||\tilde{\boldsymbol{F}}^{(tea)} - \boldsymbol{F}^{(tea)}||_2^2. \tag{6}$$

Note that the latent teacher feature $\boldsymbol{Z}^{(tea)}$ used to train diffusion model is detached and has no gradient backward from the diffusion model.

We also use a convolution layer to project the student feature to the same dimension as teacher latent feature $\boldsymbol{Z}^{(tea)}$, denoted as $\boldsymbol{Z}^{(stu)}$. It is then passed to the diffusion model to perform a reverse denoising process (Eq. (5)) and generate the denoised student feature $\hat{\boldsymbol{Z}}^{(stu)}$. Now we have the latent representation $\boldsymbol{Z}^{(tea)}$ of teacher and the denoised representation of student $\hat{\boldsymbol{Z}}^{(stu)}$, they are then used to compute the KD loss and supervise the student, *i.e.*,

$$\mathcal{L}_{\text{diffkd}} := d(\hat{\boldsymbol{Z}}^{(stu)}, \boldsymbol{Z}^{(tea)}). \tag{7}$$

Note that our DiffKD is generic and applies to various tasks (*e.g.*, classification, object detection, and semantic segmentaion) and feature types (*e.g.*, intermediate feature, classification output, and regression output). We use simple MSE loss and KL divergence loss as the distance function $d$ to compute the discrepancy of denoised student feature and teacher feature as a baseline implementation of DiffKD, while it is possible to achieve better performance with more advanced distillation losses.

### 3.3 Adaptive noise matching

As previously discussed, we treat student feature as a noisy version of teacher feature. However, the noisy level, which represents the gap between the teacher and student features, is unknown and may vary depending on different training samples. Therefore, we cannot directly determine which initial timestep we should start the diffusion process. To address this issue, we introduce an adaptive noise matching module to match the noise level of student feature to a pre-defined noise level.

As shown in Fig. 3, we construct a simple convolutional module to learn a weight $\gamma$ that fuses student output and Gaussian noise, which helps us match the student output to the same noisy level of noisy feature at initial time step $T$. Therefore, the initial noisy feature in the denoising process becomes

$$\boldsymbol{Z}_T^{(stu)} = \gamma \boldsymbol{Z}^{(stu)} + (1 - \gamma)\boldsymbol{\epsilon}_T. \tag{8}$$

This noise adaptation can be naturally optimized with the KD loss $\mathcal{L}_{\text{kd}}$, since the optimal denoised student feature that has minimal discrepancy to the teacher feature is obtained when the student feature matches the appropriate noise level in the denoising process.

**Overall loss function.** The overall loss function of DiffKD is composed of the original task loss, a diffusion loss that optimize the diffusion model, a reconstruction loss to learn the autoencoder, and a KD loss for distillation on teacher features and denoised student features, *i.e.*,

$$\mathcal{L}_{\text{train}} = \mathcal{L}_{\text{task}} + \lambda_1 \mathcal{L}_{\text{diff}} + \lambda_2 \mathcal{L}_{\text{ae}} + \lambda_3 \mathcal{L}_{\text{diffkd}}, \tag{9}$$

where $\lambda_1$, $\lambda_2$, and $\lambda_3$ are loss weights to balance the losses. We simply set $\lambda_1 = \lambda_2 = 1$ in all experiments.

Table 2: **Evaluation results of baseline settings on ImageNet.** We use ResNet-34 and ResNet-50 released by Torchvision [28] as our teacher networks, and follow the standard training strategy (B1). MSE: we implement our baseline for comparisons. †: we replace KL divergence loss with more advanced DIST loss in DiffKD.

| Student (teacher) | | Tea. | Stu. | KD [14] | Review [6] | DKD [53] | DIST [17] | MSE | DiffKD | DiffKD† |
|---|---|---|---|---|---|---|---|---|---|---|
| R18 (R34) | Top-1 | 73.31 | 69.76 | 70.66 | 71.61 | 71.70 | 72.07 | 70.58 | 72.22 | **72.49** |
| | Top-5 | 91.42 | 89.08 | 89.88 | 90.51 | 90.41 | 90.42 | 89.95 | 90.64 | **90.71** |
| MBV1 (R50) | Top-1 | 76.16 | 70.13 | 70.68 | 72.56 | 72.05 | 73.24 | 72.39 | 73.62 | **73.78** |
| | Top-5 | 92.86 | 89.49 | 90.30 | 91.00 | 91.05 | 91.12 | 90.74 | 91.34 | **91.48** |

Table 3: **Performance of students trained with strong strategies on ImageNet.** The *Swin-T* is trained with strategy B3 in Table 1, others are trained with B2. The ResNet-50 is trained by TIMM [44], and Swin-L is pretrained on ImageNet-22K.

| Teacher | Student | Top-1 ACC (%) | | | | | | |
|---|---|---|---|---|---|---|---|---|
| | | Tea. | Stu. | KD [14] | RKD [31] | SRRL [47] | DIST [17] | DiffKD |
| ResNet-50 | ResNet-34 | 80.1 | 76.8 | 77.2 | 76.6 | 76.7 | 77.8 | **78.1** |
| | MobileNetV2 | | 73.6 | 71.7 | 73.1 | 69.2 | 74.4 | **74.9** |
| | EfficientNet-B0 | | 78.0 | 77.4 | 77.5 | 77.3 | 78.6 | **78.8** |
| Swin-L | ResNet-50 | 86.3 | 78.5 | 80.0 | 78.9 | 78.6 | 80.2 | **80.5** |
| | Swin-T | | 81.3 | 81.5 | 81.2 | 81.5 | 82.3 | **82.5** |

## 4 Experiments

In this paper, to sufficiently validate the generalization of our DiffKD, we conduct extensive experiments on image classification, object detection, and semantic segmentation tasks.

### 4.1 ImageNet classification

**Settings.** Following DIST [17], we conduct experiments on baseline settings and stronger teacher settings. The training strategies for CIFAR-100 and ImageNet datasets are summarized in Tab. 1. On baseline settings, we use ResNet-18 [11] and MobileNet V1 [16] as student models, and ResNet-34 and ResNet-50 as teachers, respectively; and the training strategy (B1) is the most common one in previous methods [6, 17, 53]. While for the stronger teacher settings, we train students with much stronger teacher models (ResNet-50 and Swin-L [26]) and strategies (B2 and B3).

We implement our DiffKD on the output feature of backbone before average pooling, and the output logits of classification head, and the distance functions are MSE and KL divergence (with a temperature factor of 1), respectively. We set $\lambda_1 = \lambda_2 = \lambda_3 = 1$.

**Results on baseline settings.** We summarized the results on baseline settings in Tab. 2. Our methods outperforms existing KD methods with a large margin, especially on the MobileNet and ResNet-50 setting, DiffKD significantly surpasses previous state-of-the-art DIST [17] by $0.38\%$ on top-1 accuracy. For comparisons with our baseline one feature distillation, we also report the MSE results with the same distillation location as DiffKD. We can see that, by only adding our diffusion model for feature alignment, DiffKD obviously improves the MSE results by $1.74\%$ on ResNet-18 and $1.23\%$ on MobileNet V1. Moreover, we replace the KL divergence loss in the output logits of DiffKD with the advanced loss function DIST, which achieves further improvements. For instance, DiffKD with DIST loss improves DIST by 0.42% on ResNet-18. This indicates that our feature alignment in DiffKD is generic to different KD losses and can be further improved by changing the losses.

**Results on stronger teacher settings.** To fully investigate the efficacy of DiffKD on reducing the distillation gap between teacher feature and student feature, we further conduct experiments on much stronger teachers and training strategies following DIST. From the results summarized in Tab. 3, we can see that DiffKD outperforms DIST on all model settings, especially for the most light-weight MobileNetV2 in the table, DiffKD surpasses DIST by $0.5\%$. It is worth to remind that our DiffKD only uses the simple KL divergence loss and MSE loss on the stronger settings, the performance could be further improved if we use more advanced loss functions such as DKD [53] and DIST [17].

Table 4: **Object detection performance with baseline settings on COCO val set.** T: teacher. S: student. †: we replace MSE with an attention-based MSE loss.

| Method | AP | $AP_{50}$ | $AP_{75}$ | $AP_S$ | $AP_M$ | $AP_L$ |
|---|---|---|---|---|---|---|
| *Two-stage detectors* | | | | | | |
| T: Faster RCNN-R101 | 39.8 | 60.1 | 43.3 | 22.5 | 43.6 | 52.8 |
| S: Faster RCNN-R50 | 38.4 | 59.0 | 42.0 | 21.5 | 42.1 | 50.3 |
| Fitnet [36] | 38.9 | 59.5 | 42.4 | 21.9 | 42.2 | 51.6 |
| FRS [9] | 39.5 | 60.1 | 43.3 | 22.3 | 43.6 | 51.7 |
| FGD [48] | 40.4 | - | - | 22.8 | 44.5 | 53.5 |
| DiffKD | 40.6 | 60.9 | 43.9 | **23.0** | 44.5 | **54.0** |
| DiffKD† | **40.7** | **61.0** | **44.3** | 22.6 | **44.6** | 53.7 |
| *One-stage detectors* | | | | | | |
| T: RetinaNet-R101 | 38.9 | 58.0 | 41.5 | 21.0 | 42.8 | 52.4 |
| S: RetinaNet-R50 | 37.4 | 56.7 | 39.6 | 20.0 | 40.7 | 49.7 |
| Fitnet [36] | 37.4 | 57.1 | 40.0 | 20.8 | 40.8 | 50.9 |
| FRS [9] | 39.3 | 58.8 | 42.0 | 21.5 | 43.3 | 52.6 |
| FGD [48] | 39.6 | - | - | **22.9** | 43.7 | **53.6** |
| DiffKD | 39.7 | 58.6 | 42.1 | 21.6 | **43.8** | 53.3 |
| DiffKD† | **39.8** | **58.7** | **42.5** | 21.5 | 43.6 | 53.2 |
| *Anchor-free detectors* | | | | | | |
| T: FCOS-R101 | 40.8 | 60.0 | 44.0 | 24.2 | 44.3 | 52.4 |
| S: FCOS-R50 | 38.5 | 57.7 | 41.0 | 21.9 | 42.8 | 48.6 |
| FRS [9] | 40.9 | 60.3 | 43.6 | 25.7 | 45.2 | 51.2 |
| FGD [48] | 42.1 | - | - | **27.0** | 46.0 | 54.6 |
| DiffKD | 42.4 | 61.0 | **45.8** | 26.6 | 45.9 | 54.8 |
| DiffKD† | **42.5** | **61.1** | 45.6 | 25.2 | **46.8** | **55.1** |

Table 5: **Object detection performance with stronger teachers on COCO val set.** *CM RCNN*: Cascade Mask RCNN. †: we replace MSE with an attention-based MSE loss.

| Method | AP | $AP_{50}$ | $AP_{75}$ | $AP_S$ | $AP_M$ | $AP_L$ |
|---|---|---|---|---|---|---|
| *Two-stage detectors* | | | | | | |
| T: CM RCNN-X101 | 45.6 | 64.1 | 49.7 | 26.2 | 49.6 | 60.0 |
| S: Faster RCNN-R50 | 38.4 | 59.0 | 42.0 | 21.5 | 42.1 | 50.3 |
| COFD [13] | 38.9 | 60.1 | 42.6 | 21.8 | 42.7 | 50.7 |
| FKD [50] | 41.5 | 62.2 | 45.1 | 23.5 | 45.0 | 55.3 |
| FGD [48] | 42.0 | - | - | 23.7 | 46.4 | **55.5** |
| DiffKD | 42.2 | 62.8 | 46.0 | **24.2** | 46.6 | 55.3 |
| DiffKD† | **42.4** | **62.9** | **46.4** | 24.0 | **46.7** | 55.2 |
| *One-stage detectors* | | | | | | |
| T: RetinaNet-X101 | 41.2 | 62.1 | 45.1 | 24.0 | 45.5 | 53.5 |
| S: RetinaNet-R50 | 37.4 | 56.7 | 39.6 | 20.0 | 40.7 | 49.7 |
| COFD [13] | 37.8 | 58.3 | 41.1 | 21.6 | 41.2 | 48.3 |
| FKD [50] | 39.6 | 58.8 | 42.1 | 22.7 | 43.3 | 52.5 |
| FGD [48] | 40.4 | - | - | 23.4 | 44.7 | 54.1 |
| DiffKD | 40.7 | 60.0 | 43.2 | 22.2 | 45.0 | 55.2 |
| DiffKD† | **41.4** | **60.7** | **44.0** | 23.0 | **45.4** | **55.8** |
| *Anchor-free detectors* | | | | | | |
| T: RepPoints-X101 | 44.2 | 65.5 | 47.8 | 26.2 | 48.4 | 58.5 |
| S: RepPoints-R50 | 38.6 | 59.6 | 41.6 | 22.5 | 42.2 | 50.4 |
| FKD [50] | 40.6 | 61.7 | 43.8 | 23.4 | 44.6 | 53.0 |
| FGD [48] | 41.3 | - | - | **24.5** | 45.2 | 54.0 |
| DiffKD | 41.7 | 62.6 | 44.9 | 23.6 | 45.4 | **55.9** |
| DiffKD† | **41.9** | **62.8** | 45.0 | 24.4 | **45.7** | 55.3 |

**Results for CIFAR-100 dataset are summarized in Appendix C.**

## 4.2 Object detection

**Settings.** Following FGD [48], we conduct experiments on baseline settings and stronger teacher settings. On baseline settings, we adopt various network architectures, including two-stage detector Faster-RCNN [34], one-stage detector RetinaNet [23], and anchor-free detector FCOS [42], and use ResNet-50 [11] as student models, and ResNet-101 as teachers, respectively. The training strategy is the most common one in previous methods [9, 18, 48]. While for the stronger teacher settings, we train students with much stronger teacher models, including two-stage detector Cascade Mask RCNN [2], one-stage detector RetinaNet [23], and anchor-free detector RepPoints [49], and stronger backbone ResNeXt-101 (X101) [45].

We conduct feature distillation on the predicted feature maps, and train the student with our DiffKD loss $\mathcal{L}_{\mathrm{diffkd}}$, regression KD loss, and task loss. Note that we do not use linear autoencoder in DiffKD since the number of channels in FPN is only 256. We set $\lambda_1 = \lambda_2 = 1$. Besides using MSE loss in feature distillation, we also adopted a simple attention-based MSE loss (marked with † in Tab. 4) inspired by FGD [48] to balance the foreground and background distillations, which is acknowledged important for detection KD [10, 18, 48]. Details can be found in Appendix B.

**Results on baseline settings.** Our results compared with previous methods are summarized in Table 4. Our DiffKD can significantly improve the performance of student models over their teachers on various network architectures. For instance, DiffKD improve FCOS-R50 by 4.0 AP and surpasses FGD [48] by 0.4 AP. Besides, the attention-based MSE loss affords consistent improvements on vanilla DiffKD by $\sim 0.1$ AP.

**Results on stronger teacher settings.** We further investigate our efficacy on stronger teachers whose backbones are replaced by stronger ResNeXts [45]. As in Table 5, student detectors achieve

Table 6: **Semantic segmentation results on Cityscapes dataset.** †: trained from scratch. Other models are pretrained on ImageNet. FLOPs is measured based on an input size of $1024 \times 2048$.

| Method | Params (M) | FLOPs (G) | mIoU (%) Val | mIoU (%) Test | Method | Params (M) | FLOPs (G) | mIoU (%) Val | mIoU (%) Test |
|---|---|---|---|---|---|---|---|---|---|
| T: DeepLabV3-R101 | 61.1 | 2371.7 | 78.07 | 77.46 | T: DeepLabV3-R101 | 61.1 | 2371.7 | 78.07 | 77.46 |
| S: DeepLabV3-R18 | 13.6 | 572.0 | 74.21 | 73.45 | S: PSPNet-R18 | 12.9 | 507.4 | 72.55 | 72.29 |
| CWD [38] | 13.6 | 572.0 | 75.55 | 74.07 | CWD [38] | 12.9 | 507.4 | 74.36 | 73.57 |
| CIRKD [46] | 13.6 | 572.0 | 76.38 | 75.05 | CIRKD [46] | 12.9 | 507.4 | 74.73 | 74.05 |
| MasKD [18] | 13.6 | 572.0 | 77.00 | 75.59 | MasKD [18] | 12.9 | 507.4 | 75.34 | 74.61 |
| DiffKD | 13.6 | 572.0 | **77.78** | **76.24** | DiffKD | 12.9 | 507.4 | **75.83** | **75.61** |
| S: DeepLabV3-R18† | 13.6 | 572.0 | 65.17 | 65.47 | S: DeepLabV3-MBV2 | 3.2 | 128.9 | 73.12 | 72.36 |
| CWD [38] | 13.6 | 572.0 | 67.74 | 67.35 | CWD [38] | 3.2 | 128.9 | 74.66 | 73.25 |
| CIRKD [46] | 13.6 | 572.0 | 68.18 | 68.22 | CIRKD [46] | 3.2 | 128.9 | 75.42 | 74.03 |
| MasKD [18] | 13.6 | 572.0 | 73.95 | 73.74 | MasKD [18] | 3.2 | 128.9 | 75.26 | 74.23 |
| DiffKD | 13.6 | 572.0 | **74.45** | **74.52** | DiffKD | 3.2 | 128.9 | **75.71** | **74.96** |

more enhancements with our DiffKD, especially when with a Retina-X101 teacher, DiffKD gains a substantial improvement of 4.0 AP over the Retina-R50. Additionally, our methods outperforms existing KD methods with a large margin, and significantly surpasses FGD [48] when distilling on RetinaNet and RepPoints, by 1.0 and 0.6 AP, respectively. Moreover, comparing Table 4 with 5, we can infer that when the teacher is stronger, the benefit of DiffKD is more significant, as the discrepancy between the student and a stronger teacher is larger.

## 4.3 Semantic segmentation

**Settings.** Following CIRKD [46], we use DeepLabV3 [5] framework with ResNet-101 (R101) [11] backbone as the teacher network. While for the students, we use various frameworks (DeepLabV3 and PSPNet [54]) and backbones (ResNet-18 [11] and MobileNetV2 [37]) to valid our efficacy.

We conduct feature distillation on the predicted segmentation maps, and train the student with our DiffKD loss and task loss, as formulated in Eq. (9). Note that we do not use linear autoencoder in DiffKD since the number of channels in segmentation map is only 19. Detailed training strategies are summarized in Appendix B.

**Results.** The experimental results are summarized in Tab. 6. Our DiffKD significantly outperforms the state-of-the-art MasKD on all settings. For example, on DeepLabV3-R18 student, DiffKD improves MasKD by 0.78% on val set and 0.65 on test set.

## 4.4 Ablation study

**Effects of adopting DiffKD on different types of features.** On image classification, we distill both intermediate feature (the output feature of backbone before average pooling) and output logits with DiffKD. Here we conduct experiments to compare the effects of distillations on different features of MobileNetV1 student and ResNet-50 teacher in Tab. 7. We can see that, by adopting DiffKD to align the student features, both feature-level DiffKD and logits-level DiffKD can obtain obvious improvements without changing the loss functions. Meanwhile, combining feature and logits distillations together in our final method achieves the optimal 73.62% top-1 accuracy.

Table 7: **Comparisons of different distillation features in DiffKD.**

| Method | Top-1 (%) |
|---|---|
| w/o KD | 70.13 |
| MSE | 72.39 |
| KL div. | 70.68 |
| DiffKD (feature) | 73.16 |
| DiffKD (logits) | 72.89 |
| DiffKD (feature + logits) | **73.62** |

**Effects of linear autoencoder.** We compare different dimensions of linear autoencoder in Tab. 8. We can see that, the linear autoencoder can significantly reduces the FLOPs of diffusion model, but if we

Table 8: **Comparisons of different dimensions of linear autoencoder in DiffKD.** We report the top-1 accuracy and FLOPs of diffusion models with MobileNetV1 student and ResNet-50 teacher on ImageNet.

|  | w/o AE | 128 | 256 | 512 | 1024 | 2048 |
|---|---|---|---|---|---|---|
| Top-1 (%) | 73.47 | 72.84 | 73.22 | 73.53 | 73.62 | 73.58 |
| FLOPs (G) | 1.41 | 0.04 | 0.09 | 0.21 | 0.58 | 1.82 |

conduct high compression ratios like 128 and 256 dimensions, the distillation performance will be severely weakened. Interestingly, AE with 512, 1024, and 2048 dimensions can outperform the one without AE, a possible reason is that the AE encodes common and valuable information in the original feature for reconstruction and would have better feature representations than the original feature. As a result, we use AE with 1024 dimension in our experiments for better performance-efficiency trade-off.

**Effects of adaptive noise matching.** We propose adaptive noise matching (ANM) to match the noisy level of student feature to the correct initial level in the denoising process. Here we conduct experiments to validate its efficacy. We train MobileNetV1 student with the only removal of ANM in our final method, and the DiffKD without ANM obtains 73.34% top-1 accuracy, which has a decrease of 0.28 on our DiffKD with ANM. This indicates that ANM can improve the performance by generating better denoised student feature.

**Ablation on numbers of score function evaluations (NFEs).** In diffusion models, the number of score function evaluations (NFEs) is a important factor for controlling the generation quality and efficiency. The early method such as DDPM [15] requires to run a complete timesteps in the reverse denoising process as training, which leads to a heavy computational budget. Recently, some works [27, 40] have been proposed to accelerate the denoising process by sampling a small number of timesteps. In this paper, we use DDIM [40] for speedup. Here, we conduct experiments to show the influence of different NFEs. As shown in Fig. 4, compared to the 72.39% accuracy of MSE baseline without denoising on student feature, only one-step denoising also achieves a significant improvement. However, its performance is weaker than that of those larger NFEs due to the limitation of denoising

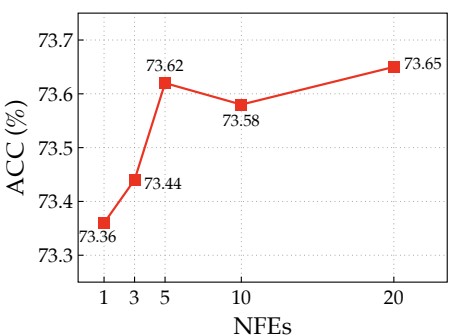

Figure 4: **Results of DiffKD with different NFEs.** We use MobileNetV1 student and ResNet-50 teacher on ImageNet.

quality. A NFEs of 5 would suffice in our KD setting to achieve promising performance, and we use it in all experiments for a better efficiency-accuracy trade-off.

## 5 Conclusion

In this paper, we investigate the discrepancy between teacher and student in knowledge distillation. To reduce the discrepancy and improve the distillation performance, we proceed from a new perspective and propose to explicitly eliminate the noises in student feature with a diffusion model. Based on this idea, we further introduce a light-weight diffusion model with a linear autoencoder to reduce the computation cost of our method, and an adaptive noise matching module to align the student feature with the correct noisy level, thus improving the denoising performance. Extensive experiments on image classification, object detection, and semantic segmentation tasks validate our efficacy and generalization.

## Acknowledgements

This work was supported in part by the Australian Research Council under Projects DP210101859 and FT230100549.

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

# A  Related Work

## A.1  Representation gap in knowledge distillation

An emerging topic in knowledge distillation is the representation gap between teacher and student. Recently, the models have been designed larger and more complicated, with remarkable performance improvements compared to the traditional models. As a result, an intuitive idea to improve efficient models is to distill them from a stronger teacher model. However, recent studies [29, 39] find that KD performance with a stronger teacher is poor and even worse than the normal teacher. TAKD [29] states that the student can only learn the knowledge effectively from a teacher model up to a fixed capacity, and proposes training multiple teaching assistants that have moderate capacities compared to the narrow student and a huge teacher. The teach assistants are trained sequentially with their former larger teach assistants, and the final smallest teach assistant is used to train the student. DAKD [39] further improves TAKD by densely connecting all models (teacher, assistant teachers, and student) together and letting the student choose the optimal teacher for each sample. SFTN [30] proposes to learn the teacher model with the supervision of student model, which obtains a student-friendly teacher with smaller representation gap to the student. However, such methods suffer from complex distillation algorithms and heavy computational cost for training the teacher model, therefore, they are not applicable in practice. More recently, DIST [17] proposes an efficient and simple approach, which aims at relaxing the exact matching in previous KL divergence loss with a correlation-based loss, which performs better when the discrepancy between teacher and student is large. However, such heuristic loss in DIST only adapts to classification outputs, and thus for dense prediction tasks such as object detection, which requires detailed semantic information in intermediate features, the efficacy is limited. Moreover, the Pearson correlation in DIST only has shift and scale invariances, whereas for the complex situation of dicrepancy in features, it cannot ignore all noises for a clean distillation.

As a result, we present DiffKD, which adapts well to multiple types of features and tasks, and handles better in eliminating the discrepancy in distillation features.

## A.2  KD in dense prediction tasks

Different from the image classification task that only needs to recognize the overall classification of the image, object detection and semantic segmentation are referred as dense prediction tasks that have to predict the bounding boxes and classes for all the objects inside the image, or the pixel-level segments of image. As a result, effectively distilling the knowledge of the teacher in these tasks is more challenging than classification.

Various methods [3, 10, 20, 38, 48, 51] are proposed to improve KD performance in object detection. Chen et al. [3] first proposes performing KD on the classification logits and regressions of the RoI head. Mimicking [20] states that the feature maps in detection model contains richer semantic information than the responses, and proposes to distill the FPN [22] features of teacher. However, distilling from the features suffers from severe imbalance of the foreground and background pixels in object detection. To address this issue, recent methods focus on selecting valuable features and propose various loss functions based on this [10, 18, 38, 48, 52].

In semantic segmentation, knowledge distillation techniques typically prioritize maintaining the structural semantic connections between the teacher and student models. To address the inconsistency between teacher and student features, He et al. [12] employ a pretrained autoencoder to optimize feature similarity in a transferred latent space. They also transfer non-local pairwise affinity maps to minimize feature discrepancies. SKD [25] conducts pairwise distillation among pixels to preserve pixel relationships and adversarial distillation on score maps to distill holistic knowledge. IFVD [43] transfers intra-class feature variation from teacher to student to establish more robust relations with class-wise prototypes. CWD [38] proposes channel-wise distillation to better mimic spatial scores along the channel dimension. CIRKD [46] proposes intra-image and cross-image relational distillations to learn better semantic relations from the teacher.

However, existing state-of-the-art KD approaches in dense prediction tasks are designed specifically for the tasks, which are difficult to be applied to various tasks, resulting in a large experiment cost in evaluating and adapting those methods. As a result, an interesting direction in KD is to generalize and unify KD methods in different tasks.

# B  Implementation Details

## B.1  Diffusion model

In all experiments, we use DDIM [40] as the noise scheduler in reverse denoising process. The total range of timesteps for training is 1000, and the initial time step for denoising is 500.

## B.2  Image classification

On ImageNet classification task, we simply set $\lambda_1 = \lambda_2 = \lambda_3 = 1$ in all experiments. (1) *Feature distillation.* For experiments that use ResNet-50 or ResNet-34 as the teacher model, we set the number of latent channels in our autoencoder to $1024$; while for Swin-L teacher, its feature for distillation has 1536 channels, so we compress it to 768 with autoencoder. (2) *Logits distillation.* We also add a DiffKD loss on the predicted classification logits. Different from the Bottleneck block used in feature distillation, we use MLP (two linear layers associated with an activation function) as an alternate of Bottleneck since the logits have only two dimensions (no spatial dimensions). Besides, the autoencoder is not used in logits distillation. On CIFAR-100 dataset, we also implement DiffKD on the output feature before average pooling layer and classification logits, but remove the autoencoder since the computation cost on CIFAR model is small.

## B.3  Object detection

For object detection task, we conduct feature distillation on the predicted feature maps, and train the student with our DiffKD loss $\mathcal{L}_{\mathrm{diffkd}}$, regression KD loss, and task loss. Note that we do not use linear autoencoder in DiffKD since the number of channels in FPN is only 256. We set $\lambda_1 = \lambda_3 = 1$. We adopt ImageNet pre-trained backbones during training following previous KD works [18]. All the models are trained with the official strategies (SGD, weight decay of 1e-4) of $2\times$ schedule in MMDetection [4]. We run all the models on 8 V100 GPUs.

**Loss weights**: We set DiffKD loss weight to $5$ and regression loss weight to $1$ on *Faster RCNN* students. For other detection frameworks, we simply adjust the loss weight of DiffKD to keep a similar amount of loss value as *Faster RCNN*. Concretely, the loss weights of $\mathcal{L}_{\mathrm{DiffKD}}$ on *RetinaNet*, *FCOS*, and *RepPoints* are 5, 5, and 15, respectively.

## B.4  Semantic segmentation

Following CIRKD [46] and MasKD [18], we train the models with standard data augmentations including random flipping, random scaling in the range of $[0.5, 2]$, and a crop with size $512\times1024$. An SGD optimizer with momentum 0.9 is adopted, and the learning rate is annealed using a polynomial scheduler with an initial value of 0.02. For DiffKD, we distill the knowledge of the segmentation predictions of teacher. Since these predictions are probabilistic distributions, we use DIST [17] as the distance function $d$ in KD loss. We do not involve autoencoder since the segmentation prediction has only 19 channels on Cityscapes dataset, and set $\lambda_1 = \lambda_3 = 1$.

# C  More Experiments

## C.1  Results on CIFAR-100 dataset

We summarize the CIFAR-100 results in Tab. 9. Our DiffKD surpasses previous methods in most cases. Moreover, comparing homogeneous architecture settings and heterogeneous architecture settings, DiffKD gains more significant improvements on heterogeneous settings compared to the standalone training results, which indicates that our method can deal better with the discrepancy between teacher and student models.

## C.2  Effect of efficient diffusion model

We train RetinaNet R50 student with R101 teacher with $1\times$ schedule in COCO, and compare our Effieicnt DM with original UNet in DDPM using DiffKD. As shown in Tab. 10, with feature shape (256, 80, 124), the original UNet has much larger parameters and GFLOPs, and thus leads to $3\times$

Table 9: **Evaluation results on CIFAR-100 dataset.** The upper and lower models denote teacher and student, respectively.

| Method | Homogeneous architecture style | | | Heterogeneous architecture style | | |
| --- | --- | --- | --- | --- | --- | --- |
| | WRN-40-2 WRN-40-1 | ResNet-56 ResNet-20 | ResNet-32x4 ResNet-8x4 | ResNet-50 MobileNetV2 | ResNet-32x4 ShuffleNetV1 | ResNet-32x4 ShuffleNetV2 |
| Teacher | 75.61 | 72.34 | 79.42 | 79.34 | 79.42 | 79.42 |
| Student | 71.98 | 69.06 | 72.50 | 64.6 | 70.5 | 71.82 |
| FitNet [36] | 72.24±0.24 | 69.21±0.36 | 73.50±0.28 | 63.16±0.47 | 73.59±0.15 | 73.54±0.22 |
| VID [1] | 73.30±0.13 | 70.38±0.14 | 73.09±0.21 | 67.57±0.28 | 73.38±0.09 | 73.40±0.17 |
| RKD [31] | 72.22±0.20 | 69.61±0.06 | 71.90±0.11 | 64.43±0.42 | 72.28±0.39 | 73.21±0.28 |
| PKT [32] | 73.45±0.19 | 70.34±0.04 | 73.64±0.18 | 66.52±0.33 | 74.10±0.25 | 74.69±0.34 |
| CRD [41] | 74.14±0.22 | 71.16±0.17 | 75.51±0.18 | 69.11±0.28 | 75.11±0.32 | 75.65±0.10 |
| KD [14] | 73.54±0.20 | 70.66±0.24 | 73.33±0.25 | 67.35±0.32 | 74.07±0.19 | 74.45±0.27 |
| DIST [17] | **74.73**±0.24 | 71.75±0.30 | 76.31±0.19 | 68.66±0.23 | 76.34±0.18 | 77.35±0.25 |
| DiffKD | 74.09±0.09 | **71.92**±0.14 | **76.72**±0.15 | **69.21**±0.27 | **76.57**±0.13 | **77.52**±0.21 |

Table 10: **Comparisons of different dimensions of linear autoencoder in DiffKD.** We report the top-1 accuracy and FLOPs of diffusion models with MobileNetV1 student and ResNet-50 teacher on ImageNet.

| | w/o AE | 128 | 256 | 512 | 1024 | 2048 |
| --- | --- | --- | --- | --- | --- | --- |
| Top-1 (%) | 73.47 | 72.84 | 73.22 | 73.53 | 73.62 | 73.58 |
| FLOPs (G) | 1.41 | 0.04 | 0.09 | 0.21 | 0.58 | 1.82 |

training time. The original UNet only achieves similar performance as our Efficient DM, as the generation of features is easier than images, and a small model would suffice.

## C.3 Effect of adaptive noise matching (ANM)

To validate the efficacy of ANM more thoroughly, we conducted further experiments in more model settings and training strategies, as summarized in Tab. 11. We can see that, when with stronger strategy and teacher, the improvement of ANM is more significant (1.2% improvement on MobileNetV2 compared to 0.3% and 0.5% improvements on MobileNetV1 and ResNet-18). One possible reason is that, when the augmentations and teacher become stronger, the noisy gaps between predicted features of teacher and student become more various, and therefore ANM is more effective in matching the noisy levels.

## C.4 Statistics of the learned noise weight $\gamma$

To analyze the effectiveness of ANM, we first show the distribution of noise weight $\gamma$ in Fig. 5 (a) of the rebuttal PDF. Revealing that the student feature is noised with $\boldsymbol{Z}_T^{(stu)} = \gamma \boldsymbol{Z}^{(stu)} + (1-\gamma)\boldsymbol{\epsilon}_T$, the larger $\gamma$ denotes smaller additional noise added.

We can see that a large amount of values is in the range of $\gamma > 0.9$, indicating that the student feature itself contains nonnegligible noises and only requires small noises to match the initial noise level, while there also exist some cleaner samples that require large noises.

We also plot the curves of average $\gamma$ in each epoch during training. Fig. 5 (b) indicates that, at the beginning of training, the student feature contains more noises, so only small weights of noises should be added. When the model gets converged, the noise in student feature becomes smaller and $\gamma$ goes smaller to match the noise level accordingly.

Table 11: **Effects of adaptive noise matching (ANM) on various model settings.**

| Student | Teacher | Strategy | w/ ANM | w/o ANM |
|---------|---------|----------|--------|---------|
| MobileNetV1 | ResNet-50 | B1 | 73.6 | 73.3 |
| ResNet-18 | ResNet-34 | B1 | 72.2 | 71.7 |
| MobileNetV2 | ResNet-50-SB | B2 | 74.9 | 73.7 |

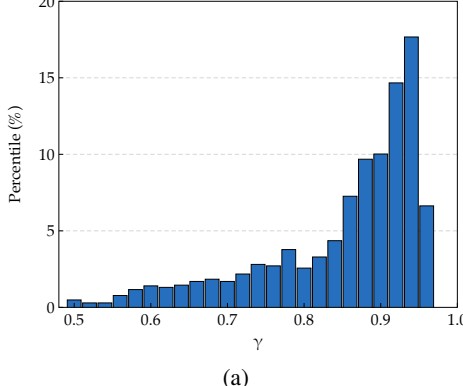

(a)

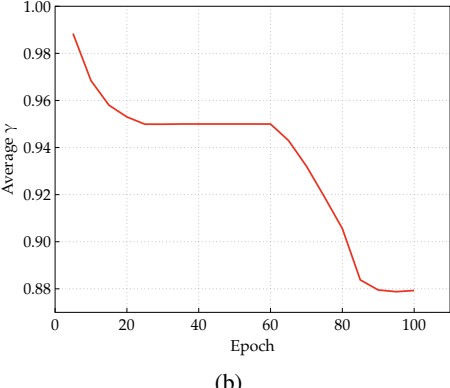

(b)

Figure 5: Statistics of learned noise weight $\gamma$ on MobileNetV1 student. (a) Histogram of learned $\gamma$ during training. (b) curves of average $\gamma$ in each epoch during training.

## C.5  Effects of different feature types of distillations on object detection

We conduct experiments on COCO dataset to compare the effects of performing distillations on FPN features, classification outputs, and regression outputs. As the results shown in Tab. 12, compared with the original KD without feature denoise in DiffKD, our method obtains significant improvements on all the feature types. Besides, comparing these features, distillation on FPN feature obtains relative high performance, which demonstrates that the semantic information in intermediate features is more valuable than the responses. We also conduct experiment to combine all the feature distillations in DiffKD together, while it achieves similar performance of FPN feature, this infers that the distillation on FPN feature is sufficient to obtain a good performance.

Table 12: **Comparison of original diffusion model (DM) and our efficient DM on COCO dataset.**

| DM | Params | GFLOPs | Training time | AP | AP | AP |
|----|--------|--------|---------------|----|----|----|
| UNet (DDPM) | 13.62 M | 650.77 | 1.80×8 GPU Days | 39.1 | 58.0 | 41.9 |
| Efficient DM (ours) | 0.21 M | 132.76 | 0.59×8 GPU Days | 39.2 | 58.1 | 42.0 |

## C.6  Efficiency analysis

In DiffKD, we use a light-weight diffusion model to denoise the student feature. Here we analyze the computation efficiency with comparisons to other methods. Compared to the vanilla KD, DiffKD has additional computations to denoise the intermediate feature and output logits on ImageNet. Specifically, for ResNet-18 student and ResNet-34 teacher, the additional FLOPs is 800M. However, compared to other feature-based KD methods, the cost is acceptable. For example, CRD [41] uses extra 260M FLOPs, ReviewKD has an addtional computation cost of 1900M.

# D  Discussion

## D.1  Limitation

In this paper, we only implement our diffusion model with simple convolutions and DDIM inference algorithm, while there exists recent advances of diffusion models that propose better transformer-based models and more efficient inference algorithm. Besides, we only use the traditional mean square error and KL divergence as our KD loss functions, while many novel losses could be leveraged

to further improve the distillation performance. The computational cost is larger than the simple logits distillation methods such as KD [14] and DIST [17], but the cost is comparable to existing feature distillation methods and does not affect the computation cost in inference.

## D.2 Societal impacts

Investigating the efficacy of the proposed method would consume considerable computing resources. These efforts can contribute to increased carbon emissions, which could raise environmental concerns. However, the proposed knowledge distillation method can improve the performance of light-weight compact models, where replacing the heavy models with light models in production could save more energy consumption, and it is necessary to validate the efficacy of DiffKD adequately.

## E   Visualization of Predicted Classification Scores

We visualize the predicted classification scores in Fig. 6. The original predictions of student, have different sharpness compared to the teacher's, while the denoised predictions align better to the predictions of teacher.

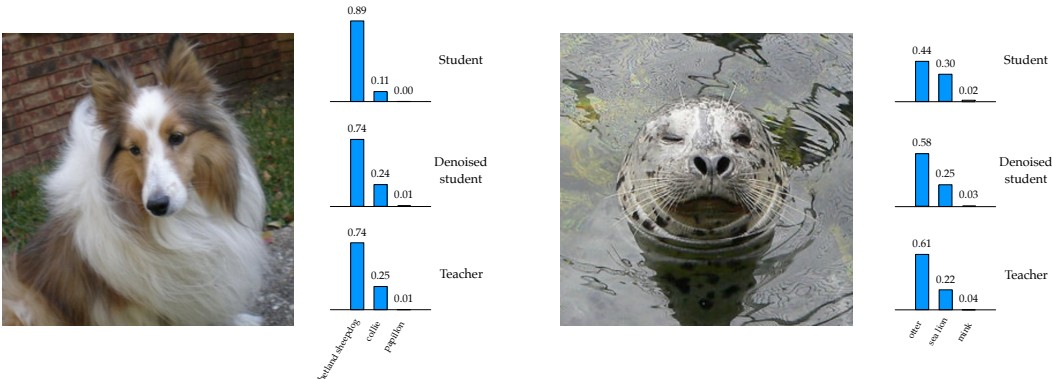

Figure 6: Visualization of predicted scores of MobileNetV1 student on ImageNet validation set. DiffKD has an effect of matching the sharpness of the probabilistic distributions of teacher and student.

## F   Visualization of Features

### F.1   Visualization details

We visualize the features of student and teacher models in the first output (stride 4) of FPN. The models used to extract the feature are RetinaNet-R50 (student) and RetinaNet-X101 (teacher) trained on COCO dataset. Following FGD [48], we average the feature map along channel axis and perform softmax on the spatial axis to measure the saliency of each pixel. Formally, with a given feature map $\boldsymbol{X} \in \mathbb{R}^{C \times HW}$, we first average the channels and get $\boldsymbol{X}' \in \mathbb{R}^{HW}$, where $X'_i = \frac{1}{HW} \sum_{i=1}^{HW} (\boldsymbol{X}_{:,i})$. Then we generate the attention map for visualization as

$$\boldsymbol{V} = H \cdot W \cdot softmax(\boldsymbol{X}'/\tau), \tag{10}$$

where $\tau$ is the temperature factor for controlling the softness of distribution, and we set $\tau = 0.5$.

### F.2   More visualizations

We visualize the student features, denoised student features, and teacher features in Fig. 7. First, by comparing the original student and teacher features, we can conclude that the discrepancy between student and teacher features is fairly large, and the student feature contains more noises and is not as salient as the teacher feature. While for the denoised feature generated by our DiffKD, it is very similar to the teacher feature, this infers that distillation on the denoised student feature and teacher feature can get rid of the noises that disturb the optimization.

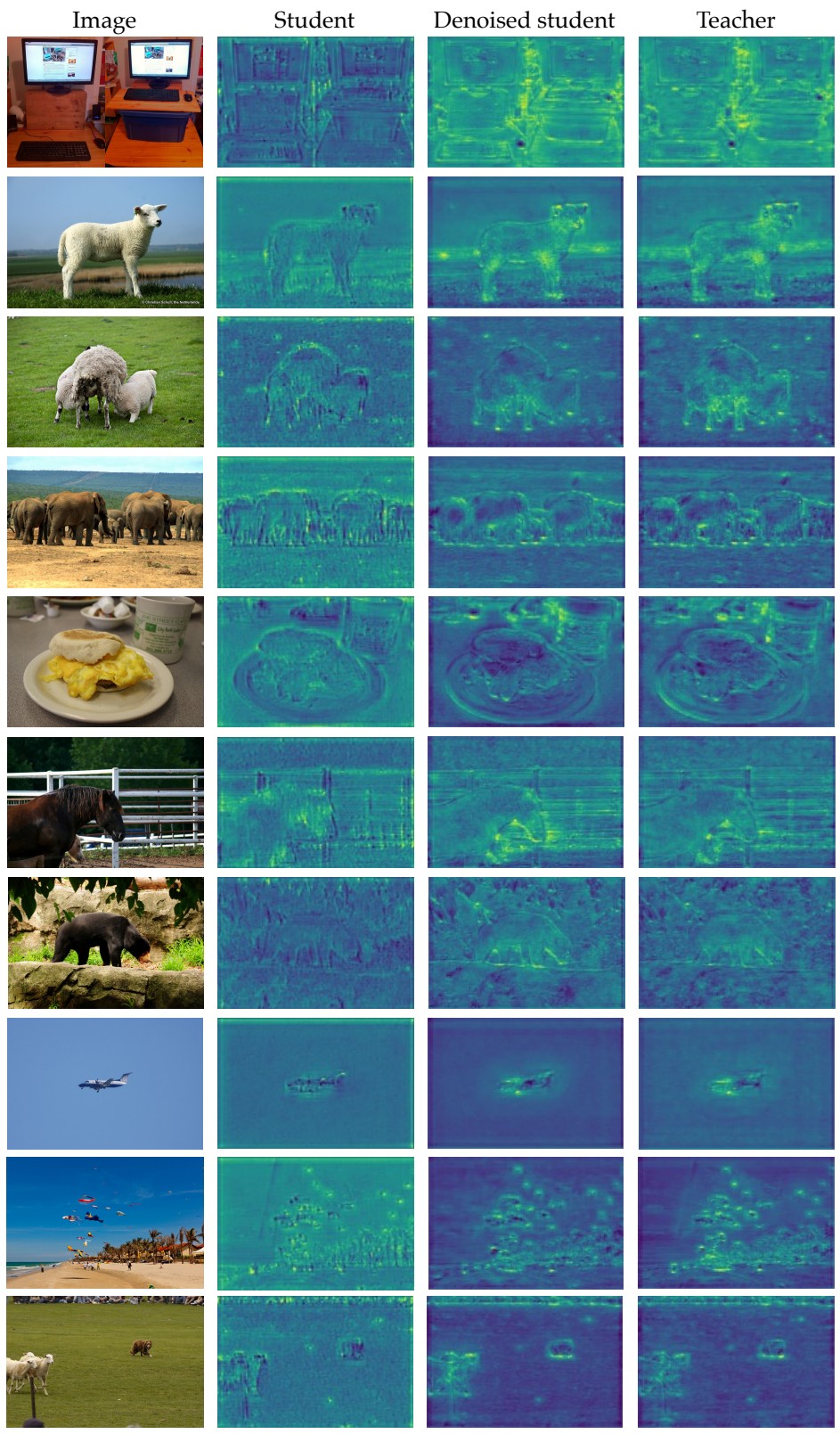

Figure 7: **Visualizations of student features, denoised student features and teacher features on COCO dataset.**

