# OpenReview forum: "Knowledge Diffusion for Distillation"
_NeurIPS.cc/2023/Conference — NeurIPS 2023 poster_

### Official Review · Reviewer_AKTb · 2023-07-03

**Soundness:** 2 fair
**Presentation:** 3 good
**Contribution:** 2 fair
**Rating:** 5
**Confidence:** 4

**Summary:**

This authors propose to explicitly eliminate the noises in student feature with a diffusion model to reduce the dicrepancy between student and teacher model for better knowledge distillation. Specifically, they build a lightweight diffusion model to reduce computation cost and introduce an adaptive noise matching module to align the student feature with the approximate noisy level of intermediate diffusion.

**Strengths:**

1.The paper adopts the diffusion model to model the teacher feature and denoise the student feature for reducing the dicrepancy between student and teacher model.
2.Quantitative experiments on multiple high-level tasks are provided to evaluate the effectiveness of the proposed method.
3.The paper is well organized and presented.

**Weaknesses:**

1.For methods,
(1)The technique contribution of this work is not significant.
(2)It is intuitive to regard the student as a noisy version of the teacher, which lacks the theoretical analysis. And the NFEs of diffusion model also depend on the capacity of student models for different noise levels.
(3)The design principle of adaptive noise matching is not explained clearly. For different student models, the final learned γ should be analyzed.
(4)It is still confused about how to determine the initial timesteps for the student model to reverse diffusion. The reverse diffusion is started from pure Gaussian noise, how to perform the inference for the feature output from the noise adapter. Would the γ near 0？
2.For experiments,
(1)For Table 2 and Table 3, it seems that the proposed method achieves marginal performance improvement for students trained with strong strategies. The reason should be further discussed and analyzed. Besides, it only obtains comparable performance on object detection task, which did not show obvious advantages than other methods.
(2)More visualized results should be provided, such as the affinity matrix.
(3)Whether other generative models, e.g., GANs, FLOW, can be adopted to model the dicrepancy between student and teacher model or not? It should be discussed and compared to demonstrate the effectiveness of diffusion model.
(4)In Table 6, Params and FLOPs are not shown completely.
(5)Since different model capacity determines the dicrepancy between student and teacher model, how about the difference of NFEs for different student models?
(6)Some important details are missing, such as the total timesteps for training the diffusion model, the initial timesteps for the student model to reverse diffusion.
3.Some writing typos, e.g.,
Line 13:  adpative -> adaptive
Line 136:  We -> we

**Questions:**

See Section Weakness

**Limitations:**

The paper is incrementally novel .

---

> ### Author Rebuttal · Authors · 2023-08-09
>
> > 1.1 The technical contribution of this work is not significant.
>
> We respectfully believe our contribution is sufficient. Directly applying diffusion models to KD is difficult and not straight-forward, and we provide effective solution on it with the following adaptations. (1) We assume the student feature is a noisy version of teacher feature so that the DM is naturally involved with a teacher-training and student-denoising manner. (2) The original diffusion model is computationally-expensive, we introduce efficient DM and autoencoder that effectively reduce the cost. (3) We design an adaptive noise matching module to solve the issue of unknown noise levels in student features.
>
> > 1.2.1 Theoretical analysis of regarding student as a noisy version of teacher.
>
> In KD, the student's objective is approximating the teacher outputs, i.e., $L_{kd} := d(F^{(s)}, F^{(t)})$. However, in practice, this approximation cannot be identical to the teacher feature ($L_{kd}=0$). This indicates that the equivalence $F^{(s)} - F^{(t)} = \delta$, with $\delta$ being an arbitrary non-zero tensor always holds. We regard the student as an additive noise model [1] where its output is equal to the addition of teacher output and noise $\delta$.
>
> For a shorter and more appropriate optimization of the approximation in KD, we want to discard the bias $\delta$ with a function $f$ so that it's possible to achieve the optima $F^{(t)} = f(F^{(s)})$, where
> $f(F^{(s)}) = F^{(s)} - \delta$.
>
> The transformation function $f$ needs to eliminate the additive noise $\delta$, which is similar to the task of image denoising. Therefore, we adopt diffusion models, a popular variant of generative models that can effectively generate clean images or features from the noisy ones.
>
> **References**
>
> [1] Hoyer, Patrik, et al. "Nonlinear causal discovery with additive noise models." NIPS 2008.
>
>
> > 1.2.2 NFEs of diffusion model depend on the capacity of student models for different noisy levels.
>
> Actually, in conventional diffusion models, all the numbers of NFEs are valid for generating images, but larger NFEs usually result in higher generation quality. Similarly, we can also use the same NFEs for different settings in DiffKD, and we empirically find that NFEs of 5 can obtain promising results for all the settings (see response 2.5).
>
> > 1.3 Analysis of the final learned $\gamma$ of different student models.
>
> We measured the average final learned $\gamma$ of different student and teacher models, as shown in the Table 1 of rebuttal PDF.
>
> We can see that, the $\gamma$ tends to be larger when the accuracies or architectures of teacher and student are more varied, as the noise in student feature is large and it needs smaller additional noise.
>
> > 1.4 Design principle of ANM & how to determine the initial timesteps for the student model.
>
> The student features would have different noise levels for different samples, which may not always match the fixed initial noise level. Therefore, we propose adaptive noise matching (ANM) to match the noise level to our defined level. The ANM is optimized with distillation loss, since the denoising effect becomes optimal when the noise level is exactly matched.
>
> > 2.1.1 Marginal performance improvements for students trained with strong strategies.
>
> The students trained with strong strategies have very competitive accuracies, so it is difficult to obtain significant improvements over them. But we still achieve 0.3%, 0.5%, and 0.3% gains on ResNet-34, MobileNetV2, and ResNet-50 compared to previous SOTA DIST, which we believe is sufficient to show our superiority.
>
> > 2.1.2 DiffKD only obtains comparable performance on object detection task.
>
> DiffKD is a general KD method that applies to various tasks. Unlike the compared methods such as FGD, which are specifically-designed for object detection task and have to use the ground-truth bounding boxes to assist the distillation, our method only uses features as input and MSE loss, yet achieves consistent improvements on all models with simple MSE loss. We think this is sufficient to show our superiority.
>
> > 2.2 More visualizations.
>
> See rebuttal PDF.
>
> > 2.3 Can other generative models be adopted to model the discrepancy?
>
> Applying diffusion model is more natural since we treat the student feature as a noisy teacher feature. Besides, we can also use other generative models to predict the denoised student feature, but they may face some issues. For example, GANs are known to suffer from a variety of issues such as non-convergence and training instability, and the autoencoder and flow-based models may have inferior performance. In contrast, DM offers desirable properties such as distribution coverage, a stationary training objective, and easy scalability, which is more friendly to our task.
>
> > 2.4 In Table 6, Params and FLOPs are not shown completely.
>
> Sorry for the confusion. Actually, the values of Params and FLOPs inside the table apply to all compared methods since the models are the same. We will refine this table.
>
> > 2.5 Different NFEs for different student models?
>
> To explore whether we should adopt different NFEs for different model settings or not, we added experiments to show the performance of various models under multiple NFEs, as summarized in Table 2 of rebuttal PDF. Note that due to the limitation of our computational resource, we only trained the EfficientNet-B0 model for 100 epochs with B1 strategy.
>
> We can see that all the three settings have relatively high performance at 5 NFEs, and their preferences on NFEs are similar. However, we do acknowledge that different NFEs for different settings or even samples could achieve better performance-efficiency trade-off, and it is a valuable direction for future increments.
>
> > 2.6 Some important implementation details are missing.
>
> The total range of timesteps for training is 1000, and the initial time step for denoising is 500. We will report more details and release our code in the publication.

---

> ### Author Response · Authors · 2023-08-12
> **Discussion to Reviewer AKTb**
>
> Dear Reviewer AKTb,
>
> We sincerely thank you for your efforts in reviewing our paper. We have provided corresponding responses and results, which we believe have covered your concerns. We hope to further discuss with you whether your concerns have been addresses or not. Please let us know if you still have any unclear part of our work.
>
> Best,
> Authors

---

> ### Author Response · Authors · 2023-08-18
> **Additional Experiments with GANs**
>
> Dear Reviewer AKTb,
>
> We extend our sincere gratitude for your dedicated effort and invaluable feedback aimed at enhancing the quality of our paper. We are truly appreciative of your contributions.
>
> We are pleased to provide you with an update on our progress, specifically concerning the incorporation of GANs in the context of KD.
>
> As the author-reviewer discussion period approaches its final days, we wish to encourage open and constructive dialogue. If you have any concerns, insights, or suggestions, we are eagerly available for discussion. Your input is highly valuable to us, and we are committed to further refining and advancing our method based on your expertise.
>
> ---
>
> **Experiments on GANs.**
>
> We conducted experiments that integrated mainstream GAN methods in lieu of our DM to transform student features. Specifically, we employed a generator designed with identical architecture to our DM. The generated refined student features were subsequently optimized using a discriminator and an adversarial loss within the GAN framework.  The discriminator is trained with refined student feature (fake data) and teacher feature (real data).
>
> For a comprehensive evaluation, we compared multiple adversarial losses commonly used in GANs, including DCGAN [1], LSGAN [2], and Hinge loss [3]. We then compared the outcomes of these GAN variants with the results of our DiffKD approach in the table below.
>
> |Method|Teacher|Student|ImageNet ACC (%)|
> |--|:--:|:--:|:--:|
> |MSE baseline|ResNet-50|MobileNetV1|72.39|
> |DCGAN|ResNet-50|MobileNetV1|72.89|
> |LSGAN|ResNet-50|MobileNetV1|72.22|
> |Hinge Loss GAN|ResNet-50|MobileNetV1|71.24|
> |DiffKD|ResNet-50|MobileNetV1|**73.62**|
>
> Our findings demonstrate that, the GAN-based methods we investigated consistently yielded inferior performance compared to DiffKD. Notably, the performance of these GAN variants exhibited sensitivity to the choice of adversarial loss. This susceptibility can be attributed to the well-recognized challenges of non-convergence and training instability associated with GANs.
>
> ---
>
> **References**
>
> [1] Radford, A., Metz, L., & Chintala, S. (2015). Unsupervised representation learning with deep convolutional generative adversarial networks. arXiv preprint arXiv:1511.06434.
> [2] Mao, X., Li, Q., Xie, H., Lau, R. Y., Wang, Z., & Paul Smolley, S. (2017). Least squares generative adversarial networks. In Proceedings of the IEEE international conference on computer vision (pp. 2794-2802).
> [3] Lim, J. H., & Ye, J. C. (2017). Geometric gan. arXiv preprint arXiv:1705.02894.

---

> > ### Comment · Reviewer_AKTb · 2023-08-20
> >
> > I appreciate the authors carefully answering my questions which covers my concerns. The motivation and contributions had better be polished further. I would  upgrade the score.

---

### Official Review · Reviewer_78h4 · 2023-07-03

**Soundness:** 4 excellent
**Presentation:** 3 good
**Contribution:** 3 good
**Rating:** 6
**Confidence:** 4

**Summary:**

This paper proposes a novel method of knowledge distillation. It uses a diffusion model to denoise the student model features, reducing the gap between  the teacher and student model. An auto-encoder is also designed to reduce the computational efforts and an adaptive noise module to improve the denoising effect. The method is extensively evaluated and achieves SOTA results in image classification, object detection, and semantic segmentation.

**Strengths:**

This paper is innovative in applying a diffusion model to knowledge distillation to reduce the gap between the teacher and the student model.
The method proposed in this paper is highly applicable and can be used for different types of features. It works well for a variety of tasks such as image classification, target detection, and semantic segmentation.

**Weaknesses:**

Some of the hyperparameters are not described clearly.
Line 177 says λ1=λ2=1, does this indicate that λ3 is set differently? line 220 says λ1=λ3=1.

Distillation methods are usually very sensitive to different backbones/downstream tasks? More evaluations and clarifications of this can be added.

**Questions:**

1. Is the training of the autoencoder and diffusion model done at the same time as the distillation training (according to Equation 9)? Does this cause inadequate training of teacher features and denoising effects at the beginning of training to interfere with student model learning?

2. Why are the methods of comparison in Tables 2 and 3 different?

---

> ### Author Rebuttal · Authors · 2023-08-09
>
> We sincerely thank the reviewer for the efforts in reviewing our paper and the positive evaluation. Our responses according to the reviewer's comments are summarized as follows.
>
> ---
>
> > 1. Hyperparameters of $\lambda_1$, $\lambda_2$, and $\lambda_3$.
>
> There is a typo in line 220, and $\lambda_3$ should be $\lambda_2$. We clarify our hyperparameters as follows. All our experiments use $\lambda_1=\lambda_2=1$. For image classification and semantic segmentation tasks, the $\lambda_3$ is also set to 1; while for object detection task, we set $\lambda_3=1$ on Faster RCNN students, while for other detection frameworks, $lambda_3$ is adjusted according to their loss values compared to Faster RCNN (details are summarized in lines 498~501 of the appendix).
>
> ---
>
> > 2. Are distillation methods sensitive to different backbones or downstream tasks?
>
> Most recent advanced distillation methods are designed specifically for one task. For example, DKD [1] and DIST [2] are designed for distilling logits on classification task, while feature-level distillation on downstream tasks such as object detection is more effective (DIST obtains 40.4 mAP on Faster RCNN-R50, while the state-of-the-art FGD [3] has 42.0 mAP). Nevertheless, FGD is sophisticatedly-designed for object detection task, and it requires the ground-truth bounding boxes to compute the loss, which cannot be directly applied to other tasks. One recent work, MGD [4], proposes a feature-level distillation loss and conducted experiments on classification, detection, and segmentation tasks, but its performance on ImageNet classification is not as competitive as that on downstream tasks (e.g., MGD has 71.58% on ResNet-18, while DIST achieves 72.07% accuracy).
>
> To this end, our work does not focus on customizing the loss function, but aims to design a general method for all the feature types and tasks and gain consistent improvements with the most common distillation losses.
>
> ---
>
> > 3. Is the training of autoencoder and diffusion model done at the same time as distillation training? Does this cause inadequate training of teacher features and denoising effect at the beginning of training?
>
> Yes, the autoencoder and diffusion model are trained simultaneously with distillation training, as we found that they converge very quickly and may have minor impact on training stability. We added experiments on ResNet-50 teacher and MobileNetV1 student to prove this. As the table shows below, we train DiffKD with pretrained and fixed AE, DM, and AE & DM, respectively, and all settings have similar accuracies.
>
> |Ours|Pretrained AE|Pretrained DM|Pretrained AE & DM|
> |:--:|:--:|:--:|:--:|
> |73.62|73.68|73.51|73.55|
>
> ---
>
> > 4. Why are the methods of comparison in Table 2 and 3 different?
>
> The settings in Table 2 are the most common benchmark for ImageNet distillation, we select the recent state-of-the-art methods and directly report the results in their papers for basic comparisons. While the settings in Table 3 are with stronger models and strategies proposed by DIST [2], we regretfully have not enough computation resources to re-implement all methods in Table 2 on these settings.
>
> ---
>
> **References**
>
> [1] Zhao, Borui, et al. "Decoupled knowledge distillation." Proceedings of the IEEE/CVF Conference on computer vision and pattern recognition. 2022.
>
> [2] Huang, Tao, et al. "Knowledge distillation from a stronger teacher." Advances in Neural Information Processing Systems 35 (2022): 33716-33727.
>
> [3] Yang, Zhendong, et al. "Focal and global knowledge distillation for detectors." Proceedings of the IEEE/CVF Conference on Computer Vision and Pattern Recognition. 2022.
>
> [4] Yang, Zhendong, et al. "Masked generative distillation." European Conference on Computer Vision. Cham: Springer Nature Switzerland, 2022.

---

> ### Author Response · Authors · 2023-08-12
> **Discussion to Reviewer 78h4**
>
> Dear Reviewer 78h4,
>
> We sincerely thank you for your efforts in reviewing our paper. We have provided corresponding responses and results, which we believe have covered your concerns. We hope to further discuss with you whether your concerns have been addresses or not. Please let us know if you still have any unclear part of our work.
>
> Best,
> Authors

---

> ### Author Response · Authors · 2023-08-20
>
> Dear Reviewer 78h4,
>
> We express our sincere gratitude for your insightful feedback and thorough evaluation of our manuscript. We have taken careful consideration of your queries and provided comprehensive responses to address each of them.
>
> We are eager to ascertain whether all your concerns have been adequately addressed. Additionally, we would appreciate your input on whether any new concerns have arisen as a result of our responses.
>
> Thank you once again for your time and expertise in reviewing our manuscript.
>
> Regards,
> Authors

---

### Official Review · Reviewer_hqhp · 2023-07-13

**Soundness:** 3 good
**Presentation:** 3 good
**Contribution:** 3 good
**Rating:** 7
**Confidence:** 4

**Summary:**

The paper introduces a new knowledge distillation (KD) method named DiffKD, which aims to bridge the representations between teacher and student features via a diffusion model. The motivation is based on the finding that the student feature is noisier than the teacher feature, and therefore diffusion models can be leveraged to denoise the student feature. Additionally, an efficient diffusion model and a noise matching module are proposed for better efficiency and accuracy. The authors conduct KD on image classification, object detection, and semantic segmentation tasks to validate the method.

**Strengths:**

- The idea of reducing the gap between teacher and student features is an emerging and important topic in KD. This paper discusses the discrepancy between teacher and student, then proposes using diffusion models to reduce the discrepancy, which is interesting and straightforward.

- It is good to see that the method can generalize to various feature types and tasks. Unlike existing methods that often focus on specific tasks and design complex loss functions, this method can be used in various tasks and achieve advanced performance with simple loss functions.

- The technical contribution is evident. Instead of directly adopting classical diffusion models, this paper introduces a lightweight architecture with an autoencoder to speed up the model and a noise-matching module to improve performance.

- The improvements are significant in image classification, detection, and segmentation tasks.


**Weaknesses:**

- The method of denoising the student feature with a teacher-trained diffusion model is a bit strange. According to my understanding, the diffusion model is a generative model. Why don't the authors directly train a diffusion model to take the student model as input and generate a feature that is similar to the teacher for distillation?

- In Table 3, compared to DIST, the improvement of DiffKD on Swin-T is not as strong as that on smaller models.

**Questions:**

- In Table 8, why is the performance of DiffKD without AE worse than DiffKD with AE of 512, 1024, 2048 channels?

**Limitations:**

Applicable.

---

> ### Author Rebuttal · Authors · 2023-08-09
>
> We sincerely thank the reviewer for the efforts in reviewing our paper and the positive evaluation. Our responses according to the reviewer's comments are summarized as follows.
>
> ---
>
> > 1. Train diffusion model with teacher feature vs. Student feature.
>
> Training the diffusion model requires the features to be stable and consistent, while the student feature changes iteratively through the training, it is inappropriate to use student feature as the generative target. We added experiments to validate our intuition.
>
> |DiffKD|Student target|
> |:--:|:--:|
> |73.62|72.78|
>
> We can see that, training diffusion models with student target causes a significant performance drop of 0.84%, which demonstrates our intuition.
>
> ---
>
> > 2. Improvement on Swin-T is not as strong as that on smaller models.
>
> The Swin-T is a strong model that can obtain high ImageNet accuracy with standalone training, so gaining improvement on it is more difficult compared to the small models with much lower accuracies. Besides, the discrepancy between Swin-T and Swin-L is also not as large as the settings with small students. Therefore, the denoising effect is also limited.
>
> ---
>
> > 3. Why is the performance without AE worse than DiffKD with AE 512, 1024, 2048 channels?
>
> We think the AE may have an effect of purifying the features, and it focuses more on the factors that help reconstruct the input features. Therefore, distillation on AE-encoded features could guide the student to learn more from valuable features.

---

> ### Author Response · Authors · 2023-08-12
> **Discussion to Reviewer hqhp**
>
> Dear Reviewer hqhp,
>
> We sincerely thank you for your efforts in reviewing our paper. We have provided corresponding responses and results, which we believe have covered your concerns. We hope to further discuss with you whether your concerns have been addresses or not. Please let us know if you still have any unclear part of our work.
>
> Best,
> Authors

---

> > ### Comment · Reviewer_hqhp · 2023-08-18
> > **Response to the authors' rebuttal**
> >
> > Thanks for the authors' responses. All my concerns have been solved. I tend to keep my initial rate and vote for accepting this paper.

---

### Official Review · Reviewer_pbtu · 2023-07-21

**Soundness:** 3 good
**Presentation:** 3 good
**Contribution:** 3 good
**Rating:** 5
**Confidence:** 4

**Summary:**

This paper presents a novel knowledge distillation (KD) approach. The difference from the existing methods lies in the computation of discrepancy between the teacher and student signals. This paper formulates it using a diffusion model and uses a denoising procedure to reconstruct the teacher features from the student features. Experiments are performed on image classification, object detection, and semantic segmentation datasets.

**Strengths:**

+ The idea of measuring the discrepancy using diffusion models is novel.
+ The paper is well-written.

**Weaknesses:**

- Although using diffusion model to model the discrepancy is a reasonable idea, this paper lacks sufficient analysis on the essential benefit of using diffusion models for this purpose. Does this paper mean that the diffusion procedure finds the shortest path in a distorted feature space (rather than the plain Euclidean space) which is better in measuring the discrepancy? If yes, are there any validations (metrics, visualizations, etc.) for the statement?

- The proposed method mixes a large number of loss terms (Eqn 9). In the ablation part, the contribution of each term is not thoroughly ablated.

- The results for image classification, object detection and semantic segmentation are mostly conducted on weak student models. It is questionable whether the method works well on strong student models (because improvement becomes more difficult).

- The improvement beyond some competitive results is not strong enough (e.g. 0.1-0.3% gain on object detection, given that the student model is relatively weak).

**Questions:**

Please address the concerns above.

**Limitations:**

Overall, I think the idea of this paper is interesting. However, there are insufficient validations on either the performance (experiments are not strong enough) or the principle. I am looking forward to further results and/or explanations.

---

> ### Author Rebuttal · Authors · 2023-08-09
>
> We sincerely thank the reviewer for the efforts in reviewing our paper and the positive evaluation. Our responses according to the reviewer's comments are summarized as follows.
>
> ---
>
> > 1. Does this paper mean that the diffusion procedure finds the shortest path in a distorted feature space (rather than the plain Euclidean space) which is better in measuring the discrepancy? If yes, are there any validations (metrics, visualizations, etc.) for the statement?
>
> Yes, the denoising yields a distorted feature space that is better for optimizing the discrepancy. Compared to simple linear transformation or other generative models such as GAN, diffusion model (DM) offers desirable properties such as distribution coverage, a stationary training objective, and easy scalability.
>
> To validate the statement, we measure the discrepancies of original student feature and denoised student feature to the teacher feature, respectively. The discrepancy metrics are mean square error (MSE) and peak signal-to-noise ratio (PSNR, higher is better). As shown in the following table, the denoised student feature has smaller discrepancy comared to the original feature, this indicates that the distillation with denoised feature has shorter path in minimizing the discrepancy.
>
> |Feat|MSE|PSNR|
> |:--:|:--:|:--:|
> |Original student|4.62|24.32|
> |Denoised student|2.87|33.99|
>
> ---
>
> > 2. The contribution of each loss term.
>
> In Eq. (9), our loss is composed of task loss, diffusion loss $L_\mathrm{diff}$, autoencoder loss $L_\mathrm{ae}$, and kd loss $L_\mathrm{diffkd}$. We added experiments to validate the contribution of loss terms by removing each loss in the equation, as shown in the following table.
>
> |w/o KD|MSE baseline|Ours (Eq. (9))|Ours w/o $\mathcal{L}_\mathrm{diff}$|Ours w/o $\mathcal{L}_\mathrm{ae}$|
> |:--:|:--:|:--:|:--:|:--:|
> |70.13|72.39|73.62|72.37|72.68|
>
> We can infer that, both the diffusion loss $L_\mathrm{diff}$ and autoencoder loss $L_\mathrm{ae}$ are important to performance.
>
> ---
>
> > 3. Performance on strong student models.
>
> We have conducted experiments for stronger student models (ResNet-50, Swin-T) on ImageNet classification in Table 3. We can see that DiffKD can improve the standalone training of ResNet-50 and Swin-T by 2.0% and 1.2%, respectively, which are significant.
>
> ---
>
> > 4. The improvement beyond some competitive results is not strong enough (e.g., COCO dataset).
>
> Our method achieves consistent improvements on all the tasks, which is sufficient to demonstrate our superiority. Besides, our method uses simple distillation losses such as KL divergence and MSE, and our improvements over these losses are fairly significant. For COCO dataset, unlike the compared methods such as FGD, which are specifically-designed for object detection task and uses the ground-truth bounding boxes to assist the distillation, our method only uses features as input and MSE loss, yet achieves consistent improvements on all models with simple MSE loss.

---

> > ### Comment · Reviewer_pbtu · 2023-08-14
> > **Post-rebuttal comments**
> >
> > I read the authors' rebuttal and other reviewers' comments.
> >
> > I think the rebuttal addressed part of my questions. The technical contribution of this paper is not so significant but sufficient to get published at NeurIPS. Two more concerns.
> >
> > - Regarding the answer to my Q2, I am a bit curious why removing $L_\mathrm{diff}$ can downgrade the results below the MSE baseline.
> > - For Q3, I meant that a stronger student model (e.g. Swin-B) shall be tested and reported. Swin-T is not acceptable.
> >
> > I choose to keep my original rating for now. If new results on Swin-B (or comparable models) are not reported, I will consider downgrading my score.

---

> > > ### Author Response · Authors · 2023-08-14
> > > **Response to post-rebuttal comments**
> > >
> > > Dear Reviewer pbtu,
> > >
> > > Thank you for your follow-up discussions. Below, I have refined my responses to your queries:
> > >
> > > > 5. Regarding the answer to my Q2, I am a bit curious why removing $L_\mathrm{diff}$ can downgrade the results below the MSE baseline.
> > >
> > > The $L_\mathrm{diff}$ term plays a crucial role in optimizing the diffusion model within our DiffKD framework. By removing $L_\mathrm{diff}$, the parameters of the diffusion model remain in a randomly initialized state. Consequently, the model cannot accurately predict the noise in the student features and may even introduce new noise to the features. This undesired denoising of features significantly degrades the performance of DiffKD, resulting in results below the MSE baseline.
> > >
> > > > 6. For Q3, I meant that a stronger student model (e.g. Swin-B) shall be tested and reported. Swin-T is not acceptable.
> > >
> > > We have initiated experiments to train Swin-B with Swin-L. Due to the larger size of these models, the experiments require a longer time to complete. We assure you that we will include the results of these experiments in a new comment as soon as they are available.

---

> > > > ### Author Response · Authors · 2023-08-17
> > > > **Experiments on Swin-B Student**
> > > >
> > > > Dear Reviewer pbtu,
> > > >
> > > > We trained the Swin-B student with Swin-L teacher using KD, DIST, and our DiffKD, respectively. As the results shown in the following table, with the stronger student Swin-B, our DiffKD can also achieve a 1.2% improvement compared to the independent training. Furthermore, when compared to other knowledge distillation (KD) methods, DiffKD shows promising results. It improves upon vanilla KD by 1% and enhances DIST by 0.4%.
> > > >
> > > > |Student|Teacher|Method|Top-1|
> > > > |:--:|:--:|:--:|:--:|
> > > > |Swin-B|-|-|83.5|
> > > > |-|Swin-L|-|86.3|
> > > > |Swin-B|Swin-L|KD|83.7|
> > > > |Swin-B|Swin-L|DIST|84.3|
> > > > |Swin-B|Swin-L|DiffKD|**84.7**|

---

> > > > > ### Author Response · Authors · 2023-08-20
> > > > >
> > > > > Dear Reviewer pbtu,
> > > > >
> > > > > We express our sincere gratitude for your insightful feedback and thorough evaluation of our manuscript. We have taken careful consideration of your queries and provided comprehensive responses to address each of them.
> > > > >
> > > > > We are eager to ascertain whether all your concerns have been adequately addressed. Additionally, we would appreciate your input on whether any new concerns have arisen as a result of our responses.
> > > > >
> > > > > Thank you once again for your time and expertise in reviewing our manuscript.
> > > > >
> > > > > Regards,
> > > > > Authors

---

> ### Author Response · Authors · 2023-08-12
> **Discussion to Reviewer pbtu**
>
> Dear Reviewer pbtu,
>
> We sincerely thank you for your efforts in reviewing our paper. We have provided corresponding responses and results, which we believe have covered your concerns. We hope to further discuss with you whether your concerns have been addresses or not. Please let us know if you still have any unclear part of our work.
>
> Best,
> Authors

---

### Official Review · Reviewer_tFex · 2023-07-22

**Soundness:** 2 fair
**Presentation:** 3 good
**Contribution:** 2 fair
**Rating:** 6
**Confidence:** 3

**Summary:**

The authors propose DiffKD, a knowledge distillation technique based on the hypothesis that the student's feature is a noisy version of the teacher's feature. Based on this assumption, they use a diffusion model to iteratively denoise the student's features before matching with the teacher. In addition, they propose to use a linear autoencoder to make diffusion more efficient to compute, and they propose a noise adapter module to predict the student's noise level.

**Strengths:**

1. Treating the student's feature as a noisy version of the teacher's feature is an interesting idea.
2. The authors conducted experiments on three tasks and showed improvements for DiffKD.

**Weaknesses:**

1. The authors did not show strong evidence for the claim that the student's features are the noisy version of the teacher's features. While Figure 2 provides a visualization, it would be best if this can be analyzed quantitatively, as it's the central hypothesis that DiffKD is based on.
2. Figure 4 shows that denoising may not be very important, since the performance is not much better even if more denoising steps are taken. This suggests that it's possible that the noise in the student is very weak, which is further supported by the experiments showing very little improvement over existing methods (less than 1 point), despite the heavy machinery of the diffusion model.

**Questions:**

1. Could you please share some statistics about the predicted noise level from the noise adapter?

**Limitations:**

Limitations are discussed in the Appendix.

---

> ### Author Rebuttal · Authors · 2023-08-09
>
> We sincerely thank the reviewer for the efforts in reviewing our paper and positive evaluation. Our responses according to the reviewer's comments are summarized as follows.
>
> ---
>
> > 1. Evidence for the claim that the student feature is the noisy version of the teacher feature.
>
> In knowledge distillation, given the student's objective of imitating the teacher's outputs, i.e., $\mathcal{L}_\mathrm{kd} := d(\boldsymbol{F}^{(s)}, \boldsymbol{F}^{(t)})$, we can infer that the student feature is an approximation of the teacher feature. However, in practice, the student's approximation cannot be identical to the teacher feature. This indicates that the equivalence $\boldsymbol{F}^{(s)} - \boldsymbol{F}^{(t)} = \delta$ with $\delta$ being an arbitrary non-zero noise always holds.
>
> For empirical analysis, we take mean square error (MSE) and peak signal-to-noise ratio (PSNR) as the metrics to evaluate the noise ratio of original student features and denoised student features w.r.t. teacher features. As shown in the following table, the original student feature has a small PSNR value, while the denoised student feature has higher PSNR comared to the original feature, indicating that the original student contains noise, and our DiffKD can effectively reduce the noise ratio.
>
> |Feat|MSE|PSNR|
> |:--:|:--:|:--:|
> |Original student|4.62|24.32|
> |Denoised student|2.87|33.99|
>
> ---
>
> > 2. Denoising may not be very important, since the performance is not much better even if more denoising steps are taken.
>
> We find that just a few denoising steps (e.g., 5) can achieve good KD performance, since generating the cleaned student feature is not as difficult as generating an image in DM.
> However, this does not mean that the denoising is unimportant. We want to clarify that our performance gain does not solely come from the denoising process of diffusion model (DM), but also because of the training strategy of DM. The training target of DM is to predict the noise of different levels (zero to full gaussian noise) of noisy features, so the DM would be more robust and accurate to denoise and align the student feature to the teacher feature, compared to directly using student feature to predict the teacher feature. As shown in the following table, we compare DiffKD with simple transformation (we use the same architecture of DM to transform the student feature and compute distillation loss on transformed feature, instead of using the denoising process in DM). The results show that DiffKD significantly outperforms transformation even when they have the same FLOPs (NFEs=1).
>
> |DiffKD (NFEs=5)|DiffKD (NFEs=1)|Transformation|
> |:--:|:--:|:--:|
> |73.62|73.36|72.33|
>
> ---
>
> > 3. Statistics about the predicted noise level from the noise adapter.
>
> We first show the distribution of noise weight $\gamma$ in Fig. 1 (a) of the rebuttal PDF. Revealing that the student feature is noised with $\boldsymbol{Z}^{(stu)}_T = \gamma\boldsymbol{Z}^{(stu)} + (1 - \gamma)\boldsymbol{\epsilon}_T,$ the larger $\gamma$ denotes smaller additional noise added.
>
> We can see that a large amount of values is in the range of $\gamma > 0.9$, indicating that the student feature itself contains non-negligible noises and only requires small noises to match the initial noise level, while there also exist some cleaner samples that require large noises.
>
> We also plot the curves of average $\gamma$ in each epoch during training. The Fig. 1 (b) indicates that, at the beginning of training, the student feature contains more noises, so only small weights of noises should be added. When the model gets converged, the noise in student feature becomes smaller and $\gamma$ goes smaller to match the noise level accordingly.

---

> > ### Comment · Reviewer_tFex · 2023-08-16
> > **Reply to the authors**
> >
> > Firstly, thank you for the detailed reply. I really appreciate it!
> >
> > 1. I think you are right that for any two features, we can say $\boldsymbol{F}^{(s)} - \boldsymbol{F}^{(t)} = \delta$. However, I feel this is different from whether $\delta$ is truly a random Gaussian noise. I think this is also not something that MSE and PSNR can show.
> >
> > 2. The result shows one step diffusion already achieves good performance. This makes me question whether diffusion is truly necessary. On the other hand, a simple linear transformation may be too simple. Do you know how this performs if you apply GAN, for example?
> >
> > 3. I could not find the updated figure about the distribution of the noise levels. I'm sorry if I was looking at the wrong place.
> >
> > Again, thanks for the response!

---

> > > ### Author Response · Authors · 2023-08-17
> > >
> > > Thank you for your kind reply and follow-up discussion. Our responses to your queries is as follows.
> > >
> > > > 4. I feel this is different from whether $\delta$ is truly a random Gaussian noise.
> > >
> > > We want to clarify that the noise $\delta$ in our diffusion models is not limited to being a Gaussian noise. In fact, it is a mixture of Gaussian noises, which has been shown to have the capability to approximate almost all distributions [1, 2]. This choice allows us to effectively model and denoise complex data distributions.
> > >
> > > To provide further clarity, in the forward diffusion process, "_the diffusion process is fixed to a Markov chain that gradually adds Gaussian noise to the data according to a variance schedule $\beta_1,...,\beta_T$_" (DDPM [3]). In the diffusion process, it recursively predicts Gaussian condictions for multiple timesteps (it is the same as the timesteps of diffusion process in original DM).
> > >
> > > Therefore, by utilizing DMs, we can effectively eliminate any distribution of $\delta$ and generate a clean, denoised student feature that aligns with the teacher feature.
> > >
> > > > 5. The result shows one step diffusion already achieves good performance. This makes me question whether diffusion is truly necessary. On the other hand, a simple linear transformation may be too simple. Do you know how this performs if you apply GAN, for example?
> > >
> > > We apologize for any confusion caused by our previous response. To clarify, the `transformation` mentioned in our table is not a simple linear transformation. Instead, we adopt the same architecture as the diffusion model to transform the student feature. By making this comparison, we aim to demonstrate that the training manner of diffusion models provides a benefit to the distillation performance, even without the multistep denoising process.
> > >
> > > Furthermore, we have included results for different NFEs on other models in Table 2 of the rebuttal PDF. These results indicate that with ResNet-18 and EfficientNet-B0 students, the improvement achieved with 3 NFEs compared to 1 NFE is significant.
> > >
> > > Regarding your suggestion of using GANs for transformation, we have initiated an experiment to transform the student feature using a GAN. We are currently in the process of training and evaluating the results. Once the training is completed, we will report the outcomes and include them in our paper.
> > >
> > > > 6. I could not find the updated figure about the distribution of the noise levels. I'm sorry if I was looking at the wrong place.
> > >
> > > In accordance with the guidelines for this year's NeurIPS conference, authors are not allowed to update the submitted manuscript or appendix directly. Instead, we have provided a rebuttal PDF that includes additional information and clarifications. You can find the rebuttal PDF in our common response to all the reviewers, titled "Author Rebuttal by Authors," marked with an orange label.
> > >
> > > In the rebuttal PDF, Figure 1 presents the statistics of the noisy levels ($\gamma$) that you were looking for.
> > >
> > > ---
> > > **References**
> > >
> > > [1] Bishop, C. M. (2006). Pattern Recognition and Machine Learning. Springer. Chapter 9
> > > [2] McLachlan, G. J., & Krishnan, T. (2007). The EM Algorithm and Extensions. Wiley. Chapter 8
> > > [3] Ho, J., Jain, A., & Abbeel, P. (2020). Denoising diffusion probabilistic models. Advances in neural information processing systems, 33, 6840-6851.

---

> > > > ### Author Response · Authors · 2023-08-18
> > > > **Additional Experiments of Applying GANs**
> > > >
> > > > Dear Reviewer tFex,
> > > >
> > > > We conducted experiments that integrated mainstream GAN methods in lieu of our DM to transform student features. Specifically, we employed a generator designed with identical architecture to our DM. The generated refined student features were subsequently optimized using a discriminator and an adversarial loss within the GAN framework.  The discriminator is trained with refined student feature (fake data) and teacher feature (real data).
> > > >
> > > > For a comprehensive evaluation, we compared multiple adversarial losses commonly used in GANs, including DCGAN [1], LSGAN [2], and Hinge loss [3]. We then compared the outcomes of these GAN variants with the results of our DiffKD approach in the table below.
> > > >
> > > > |Method|Teacher|Student|ImageNet ACC (%)|
> > > > |--|:--:|:--:|:--:|
> > > > |MSE baseline|ResNet-50|MobileNetV1|72.39|
> > > > |DCGAN|ResNet-50|MobileNetV1|72.89|
> > > > |LSGAN|ResNet-50|MobileNetV1|72.22|
> > > > |Hinge Loss GAN|ResNet-50|MobileNetV1|71.24|
> > > > |DiffKD|ResNet-50|MobileNetV1|**73.62**|
> > > >
> > > > Our findings demonstrate that, the GAN-based methods we investigated consistently yielded inferior performance compared to DiffKD. Notably, the performance of these GAN variants exhibited sensitivity to the choice of adversarial loss. This susceptibility can be attributed to the well-recognized challenges of non-convergence and training instability associated with GANs.
> > > >
> > > > ---
> > > >
> > > > **References**
> > > >
> > > > [1] Radford, A., Metz, L., & Chintala, S. (2015). Unsupervised representation learning with deep convolutional generative adversarial networks. arXiv preprint arXiv:1511.06434.
> > > > [2] Mao, X., Li, Q., Xie, H., Lau, R. Y., Wang, Z., & Paul Smolley, S. (2017). Least squares generative adversarial networks. In Proceedings of the IEEE international conference on computer vision (pp. 2794-2802).
> > > > [3] Lim, J. H., & Ye, J. C. (2017). Geometric gan. arXiv preprint arXiv:1705.02894.

---

> > > > > ### Comment · Reviewer_tFex · 2023-08-18
> > > > >
> > > > > Thank you for the detailed updates!
> > > > >
> > > > > I feel the fact that DMs "have the capability to approximate almost all distributions" weakens the central claim of the paper that student features are noisy versions of the teachers. This feels like DM is only used as a bridge between the two features, and diffusion models are not particularly reasonable in the sense of denoising.
> > > > >
> > > > > However, I feel the results still suggest that DM can be a powerful model to align student and teacher features. Therefore, I would like to maintain my positive score for the submission.

---

> ### Author Response · Authors · 2023-08-12
> **Discussion to Reviewer tFex**
>
> Dear Reviewer tFex,
>
> We sincerely thank you for your efforts in reviewing our paper. We have provided corresponding responses and results, which we believe have covered your concerns. We hope to further discuss with you whether your concerns have been addresses or not. Please let us know if you still have any unclear part of our work.
>
> Best,
> Authors

---

### Official Review · Reviewer_NwaZ · 2023-07-24

**Soundness:** 3 good
**Presentation:** 3 good
**Contribution:** 3 good
**Rating:** 5
**Confidence:** 4

**Summary:**

This work proposes the use of the diffusion model for denoising the noisy feature of student.
It tackles the issues of employing the diffusion model for knowledge distillation which are heavy computation and inexact noisy level of student feature.
To tackle the issues, it proposes a light-weight diffusion model consisting of two bottleneck blocks in ResNet and adopt a linear auto encoder to compress the teacher feature. It also proposes an adaptive noise matching module which measures the noisy level of each student feature adaptively and specifies a corresponding Gaussian noise to the feature to match the correct noisy level in initialization.

**Strengths:**

The proposed idea is simple.

Showed what is the problem of directly applying the diffusion model to knowledge distillation. (Expensive Computation Cost, Inexact noisy level of student feature)

It has conducted various experiments on various vision tasks to show that the proposed method can be widely applicable. (classification, object detection, semantic segmentation).

It shows good performance gain on several benchmarks.

**Weaknesses:**

Applying diffusion model to denoise the student feature seems somewhat obvious approach, in other words, not novel.
The fact that the feature of student is noisier than that of the teacher is already well studied by other papers as mentioned in line 109-117.
In my opinion, the contribution of this paper comes from showing what are the practical problems of applying diffusion process to KD.
It showed two problems (Expensive Computation Cost, Inexact noisy level of student feature).
However, I am not sure if this paper has done enough experiments to show that it solved the above problems.
There are not enough analysis to prove the proposed method is efficient. (Table 8 and Figure 4 seems to be the only ones).

How is the diffusion process applied on the logit-level?

The effect of ANM is not really convincing, there is only a marginal performance gain. It would be more convincing how the ANM is helping to denoise the student feature better by analyzing the student feature with and without the noise adapter. Visualization of student feature with/without ANM could help to better understand the effect of ANM.



**Questions:**

Please refer to the Weaknesses.

**Limitations:**

Please refer to the Weaknesses.

---

> ### Author Rebuttal · Authors · 2023-08-09
>
> We sincerely thank the reviewer for the efforts in reviewing our paper. Our responses according to the reviewer's comments are summarized as follows.
>
> ---
>
> > 1. Applying diffusion model to denoise the student feature is not novel.
>
> We summarize our novelties as follows. (1) The existing methods solve the discrepancy between teacher and student by improving the loss functions or training strategies, while we are the first work that uses a new perspective of distilling valuable knowledge from the teacher by eliminating the noise in the student feature. (2) There is no practice on how to involve diffusion model (DM) in KD before. We assume the student feature is a noisy version of teacher feature so that the DM is naturally involved with a teacher-training and student-denoising manner. DM enjoys better generation performance than the other transformation modules in previous methods. (3) The original diffusion model follows a heavy UNet architecture which is computationally-expensive and heavily slows down the training speed of KD, we introduce efficient DM and autoencoder that effectively reduce the computation cost. (4) We design an adaptive noise matching module to solve the issue of unknown noise levels in student features.
>
> ---
>
> > 2. Experiments to show the efficacies of efficient model and ANM.
>
> **(1) Efficacy of efficient diffusion model.**
>
> We train RetinaNet R50 student with R101 teacher with 1x schedule in COCO, and compare our Effieicnt DM with original UNet in DDPM using DiffKD. As shown in the following table, with feature shape (256, 80, 124), the original UNet has much larger parameters and GFLOPs, and thus leads to ~3x training time. The original UNet only achieves similar performance as our Efficient DM, as the generation of features is easier than images, and a small model would suffice.
>
> |DM|Params|GFLOPs|Training time|AP|AP50|AP75|
> |:--:|:--:|:--:|:--:|:--:|:--:|:--:|
> |UNet (DDPM)|13.62 M|650.77|1.80*8 GPU Days|39.1|58.0|41.9|
> |Efficient DM (Ours)|0.21 M|132.76|0.59*8 GPU Days|39.2|58.1|42.0|
>
> **(2) Efficacy of ANM.**
>
> Actually, for the simple training strategy B1, the 0.28 performance gain of ANM on MobileNetV1 is not marginal since the 73.6% accuracy is high w.r.t to its capacity and it is challenging to obtain significant increment on the simple 100-epoch training strategy. To validate the efficacy of ANM more sufficiently, we further conducted experiments on more model settings and training strategies, as summarized in the following table.
>
> |Student|Teacher|Strategy|w/ ANM|w/o ANM|
> |:--:|:--:|:--:|:--:|:--:|
> |MobileNetV1|ResNet-50|B1|73.6|73.3|
> |ResNet-18|ResNet-34|B1|72.2|71.7|
> |MobileNetV2|ResNet-50-SB|B2|74.9|73.7|
>
> We can see that, when with stronger strategy and teacher, the improvement of ANM is more significant (1.2% improvement on MobileNetV2 compared to 0.3% and 0.5% improvements on MobileNetV1 and ResNet-18). One possible reason is that, when the augmentations and teacher become stronger, the noisy gaps between predicted features of teacher and student become more various, and therefore ANM is more effective in matching the noisy levels.
>
> ---
>
> > 3. How is the diffusion process applied on the logit-level?
>
> We have elaborated the implementation details and comparison of DiffKD on feature level and logits level in lines 481~488 of our supplementary material. The predicted logits can also be regarded as a feature that has only one dimension compared to the intermediate feature of 3 dimensions (height, width, and channels), and we can also use diffusion model (DM) to denoise the student logits feature. To achieve this, we replace the convolutional networks of our DM with MLP network, and the goal of DM is to predict the 1-D noise of the logits.
>
> ---
>
> > 4. The performance gain of ANM is marginal.
>
> See response 2 (2).
>
> ---
>
> > 5. Analyzing the student feature with or without ANM.
>
> **(1) Statistics of learned noise weight $\gamma$.**
>
> To analyze the effectiveness of ANM, we first show the distribution of noise weight $\gamma$ in Fig. 1 (a) of the rebuttal PDF. Revealing that the student feature is noised with $\boldsymbol{Z}^{(stu)}_T = \gamma\boldsymbol{Z}^{(stu)} + (1 - \gamma)\boldsymbol{\epsilon}_T,$ the larger $\gamma$ denotes smaller additional noise added.
>
> We can see that a large amount of values is in the range of $\gamma > 0.9$, indicating that the student feature itself contains non-negligible noises and only requires small noises to match the initial noise level, while there also exist some cleaner samples that require large noises.
>
> We also plot the curves of average $\gamma$ in each epoch during training. The Fig. 1 (b) indicates that, at the beginning of training, the student feature contains more noises, so only small weights of noises should be added. When the model gets converged, the noise in student feature becomes smaller and $\gamma$ goes smaller to match the noise level accordingly.
>
> **(2) Comparison of features with or without ANM.**
>
> To validate how much ANM can improve the denoised student features, we measure the mean distance between teacher (ResNet-50) features and denoised student (MobileNetV1) features with or without ANM on ImageNet validation set, as summarized in the following table.
>
> |Type|Metric|Distance w/ ANM|Distance w/o ANM|
> |:--:|:--:|:--:|:--:|
> |Intermediate feature|MSE|2.87|3.29|
> |Logits|KL div.|0.21|0.35|
>
> We can infer that ANM effectively reduces the discrepancy between teacher and student features, and therefore leads to better distillation performance.

---

> ### Author Response · Authors · 2023-08-12
> **Discussion to Reviewer NwaZ**
>
> Dear Reviewer NwaZ,
>
> We sincerely thank you for your efforts in reviewing our paper. We have provided corresponding responses and results, which we believe have covered your concerns. We hope to further discuss with you whether your concerns have been addresses or not. Please let us know if you still have any unclear part of our work.
>
> Best,
> Authors

---

> ### Comment · Reviewer_NwaZ · 2023-08-16
>
> Thank you for your kind responses.
> Some of my concerns are now a bit resolved.
>
> Another question is raised while reading your response and the paper again.
> I understood that minimizing eq.(7) would not only optimize the parameters of student but also gamma of ANM, but does it also affect the diffusion model during training?
> Is DM only trained by eq.(2) or is it also affected by eq.(7) ?
>
> Thank you.

---

> > ### Author Response · Authors · 2023-08-16
> >
> > Thank you for your kind reply. Below, we have refined my responses to your queries:
> >
> > > 6. I understood that minimizing eq.(7) would not only optimize the parameters of student but also gamma of ANM, but does it also affect the diffusion model during training? Is DM only trained by eq.(2) or is it also affected by eq.(7) ?
> >
> > In our early experiments, we have explored both options:
> > * (a) updating DM with teacher feature (eq. (4)) and distillation loss (eq. (7));
> > * (b) updating DM with only the teacher feature (eq. (4)), while discarding the gradients produced by the distillation loss (eq. (7)).
> >
> > Interestingly, we observed that both options (a) and (b) yielded almost identical performance. However, for the sake of simplicity and efficiency in implementation, we have chosen option (a) as the approach to be used in our final code.
> >
> > |Option|Teacher|Student|ACC (%)|
> > |:--:|:--:|:--:|:--:|
> > |(a)|ResNet-34|ResNet-18|72.22|
> > |(b)|ResNet-34|ResNet-18|72.18|
> > |(a)|ResNet-50|MobileNetV1|73.62|
> > |(b)|ResNet-50|MobileNetV1|73.68|
> >
> > Thanks,
> > Authors

---

> ### Comment · Reviewer_NwaZ · 2023-08-17
>
> After reading authors' rebuttal, some of my concerns are resolved, so I changed my rating to  'Borderline accept'.

---

### Author Rebuttal · Authors · 2023-08-09

Dear Reviewers,


We thank all the reviewers for their valuable comments and efforts in reviewing our paper.

We are delighted to see that Reviewer tFex, pbtu, hqhp, and 78h4 stated that our method is interesting and novel; Reviewer hqhp and 78h4 acknowledged that our method is widely applicable and has evident technical contribution.

We have also responded all the reviewers' concerns such as the effectivenesses of proposed modules, evidence of our statements, and significance of improvements with additional experiments, visualizations, and explanations.

The rebuttal PDF containing new figures and tables is attached in this comment for your reading.

Regards,

Authors

---

### Author Response · Authors · 2023-08-15

Dear Reviewers,

We wanted to kindly remind you that the author-reviewer discussion period is nearing its halfway mark. We would like to take this opportunity to ensure that our responses have adequately addressed your concerns and to inquire if there are any further questions or clarifications you may require.

Your valuable input and expertise are essential in enhancing the quality and impact of our research. We greatly appreciate the time and effort you have already dedicated to reviewing our paper, and we eagerly await your feedback.

Thank you for your attention and consideration.

Best regards,
Authors

---

### Decision · Program_Chairs · 2023-09-21

**Decision:**

Accept (poster)

**Comment:**

This work presents a method to denoise features in a student-teacher model using diffusion process.

The work is found novel with good results.

All reviewers recommended acceptance.